# Autonomous and non-cell autonomous role of cilia in structural birth defects in mice

Richard J. B. Francis[ID][1,2], Jovenal T. San Agustin[3], Heather L. Szabo Rogers[1,4], Cheng Cui[1], Julie A. Jonassen[5], Thibaut Eguether[3], John A. Follit[3], Cecilia W. Lo[ID][1]*, Gregory J. Pazour[ID][3]*

1 Department of Developmental Biology, University of Pittsburgh, Rangos Research Center, Pittsburgh, Pennsylvania, United States of America, 2 Discipline of Biomedical Sciences and Molecular Biology; College of Public Health, Medical and Veterinary Science, James Cook University, Townsville, Australia, 3 Program in Molecular Medicine, University of Massachusetts Chan Medical School, Worcester, Massachusetts, United States of America, 4 Center for Craniofacial Regeneration, Department of Oral Biology, School of Dental Medicine, University of Pittsburgh, Pittsburgh, Pennsylvania, United States of America, 5 Department of Microbiology and Physiological Systems, University of Massachusetts Chan Medical School, Worcester, Massachusetts, United States of America

* cel36@pitt.edu (CWL); gregory.pazour@umassmed.edu (GJP)

**Data Availability Statement:** All relevant data are within the paper and its Supporting Information files.

## Abstract

Ciliopathies are associated with wide spectrum of structural birth defects (SBDs), indicating important roles for cilia in development. Here, we provide novel insights into the temporospatial requirement for cilia in SBDs arising from deficiency in *Ift140*, an intraflagellar transport (IFT) protein regulating ciliogenesis. *Ift140*-deficient mice exhibit cilia defects accompanied by wide spectrum of SBDs including macrostomia (craniofacial defects), exencephaly, body wall defects, tracheoesophageal fistula (TEF), randomized heart looping, congenital heart defects (CHDs), lung hypoplasia, renal anomalies, and polydactyly. Tamoxifen inducible *CAGGCre-ER* deletion of a floxed *Ift140* allele between E5.5 to 9.5 revealed early requirement for *Ift140* in left-right heart looping regulation, mid to late requirement for cardiac outflow septation and alignment, and late requirement for craniofacial development and body wall closure. Surprisingly, CHD were not observed with 4 Cre drivers targeting different lineages essential for heart development, but craniofacial defects and omphalocele were observed with *Wnt1-Cre* targeting neural crest and *Tbx18-Cre* targeting epicardial lineage and rostral sclerotome through which trunk neural crest cells migrate. These findings revealed cell autonomous role of cilia in cranial/trunk neural crest-mediated craniofacial and body wall closure defects, while non-cell autonomous multi-lineage interactions underlie CHD pathogenesis, revealing unexpected developmental complexity for CHD associated with ciliopathies.

## Introduction

Cilia dysfunction underlies a large group of heritable human diseases referred to collectively as ciliopathies [1]. They are often associated with structural birth defects (SBDs), likely a reflection of the many roles of cilia in mediating signal transduction pathways important in

**Funding:** This work was supported by grants from the National Institutes of Health (https://grants.nih.gov/) under project numbers: GM060992 (GJP), U01HL098180 (CWL, GJP) and R01HL157103 (CWL). The funders had no role in study design, data collection and analysis, decision to publish, or preparation of the manuscript.

**Competing interests:** The authors have declared that no competing interests exist.

**Abbreviations:** AVSD, atrioventricular septal defect; CHD, congenital heart defect; DORV, double outlet right ventricle; ECM, episcopic confocal microscopy; ENU, ethylnitrosourea; Hh, hedgehog; IFT, intraflagellar transport; KO, knockout; MAPCA, major aortopulmonary collateral artery; MEF, mouse embryonic fibroblast; OA, overriding aorta; OFT, outflow tract; PTA, persistent truncus arteriosus; RAA, right-sided aortic arch; SBD, structural birth defect; SD, standard deviation; SHF, second heart field; SRP, short rib polydactyly; SRTD, short rib thoracic dysplasia; Shh, sonic hedgehog; TEF, Tracheoesophageal fistula; VSD, ventricular septal defect.

developmental processes, such as specification of the left-right body axis, limb and skeletal patterning, and development of the eye, heart, kidney, brain, lung, and other organs. Hedgehog (Hh) signaling is one of the first cell signaling pathways shown to be organized around the primary cilium [2]. It plays an important role in many developmental processes such as patterning of the neural tube, anterior-posterior patterning of the limb bud, and development of the kidney. During hedgehog signaling, dynamic changes are observed in cilia localization of the receptors Patched1 (Ptc1) and Smoothened (Smo), and the Gli transcription factors [3–5]. Cilia defects typically lead to attenuation of hedgehog signaling that can result in neural tube dorsoventral patterning defects and defects in anterior-posterior patterning of the limb bud causing polydactyly [2,5].

Cilia are also known to regulate noncanonical and canonical Wnt signaling, with cilia-regulated Wnt signaling in the embryonic node playing an important role in left-right patterning [6]. Moreover, motile and primary cilia at the node are required for generating flow and mediating mechanosensation, respectively, important for specification of the left-right body axis [7]. Hence, cilia defects can lead to abnormal organ situs specification, such as situs inversus totalis with mirror symmetric reversal of visceral organ situs, or heterotaxy with incomplete reversal or randomization of the left-right body axis [8]. As left-right asymmetry of the heart is essential for efficient oxygenation of blood, heterotaxy is often associated with severe congenital heart defects (CHDs) [7,8].

In ciliopathies, mutations affecting different components of the complex machinery mediating ciliogenesis can disrupt cilia structure and function. Cilia are assembled and maintained via the intraflagellar transport (IFT) system that provides bidirectional movement of large protein complexes called IFT particles. These IFT particles comprising IFT-A, IFT-B, and BBSomes are highly conserved, with orthologs found from the unicellular algae *Chlamydomonas* to mouse and man [9]. The IFT particles translocate cargo along axonemal microtubules using kinesin-2 and dynein-2 motors in the anterograde and retrograde directions, respectively [10]. In humans, mutations in IFT-A components have been found in ciliopathies marked by skeletal dysplasias, renal and liver abnormalities, vision defects, and other SBDs. To further investigate how cilia disruption may lead to SBDs, we conducted detailed phenotyping for SBDs in mice with mutations in the IFT-A subunit *Ift140* and created an *Ift140* floxed allele to investigate the temporospatial requirement for *Ift140* relative to these SBD phenotypes.

*IFT140* mutations (MIM 614620) are clinically associated with skeletal ciliopathies known as short rib thoracic dysplasia (SRTD) including Mainzer–Saldino syndrome (MIM 266920), asphyxiating thoracic dystrophy/Jeune syndrome (MIM 208500), and Sensenbrenner syndrome (MIM 614620) [11–17]. The diagnostic feature includes skeletal dysplasia with variably shortened ribs, a narrow trunk, shortened limbs with or without polydactyly. Other SBDs also may be observed involving the brain, retina, heart, and gastrointestinal tract. Additionally, cystic kidneys has been observed [18,19], with *IFT140* variants being the third most frequent cause of autosomal dominant polycystic kidney disease after *PKD1* and *PKD2* [20,21]. Previous report of an *Ift140* mutant mouse, *Cauli*, showed embryonic lethality at E13 with exencephaly, craniofacial defects, anophthalmia, and polydactyly [22], but mid-gestation lethality precluded analysis for renal anomalies or the thoracic dystrophy. We previously generated a floxed allele of *Ift140* and showed mice exhibited severe postnatal cystic kidney disease with *Ift140* deletion targeting the kidney collecting ducts [23] and also retinal degeneration with deletion in photoreceptor cells [24].

In this study, we investigated mice with an *Ift140* mutant allele that is viable to term and an embryonic lethal *Ift140* KO allele genetically null for *Ift140*. This analysis uncovered SBD phenotypes not previously reported in the *Cauli* mutant, including observations of small chest, renal anomalies, body wall closure defects, left-right patterning defects and CHDs. Using the

floxed allele of *Ift140*, we investigated the temporospatial requirement for *Ift140*, yielding new insights into the development requirements for cilia and unexpected complexity in cilia regulation of heart development and CHD pathogenesis.

## Results

### *Ift140* splicing mutation causes neonatal lethality and severe structural birth defects

A mutant mouse line, 220, harboring an *Ift140* splicing defect mutation was previously recovered from an ethylnitrosourea (ENU) mutagenesis screen and found to be homozygous neonatal lethal with a wide spectrum of SBDs (Table 1 and Fig 1) [25,26]. The neonatal lethality is in sharp contrast to the prenatal lethality seen in most other IFT mutant mice [2,27], including the *Cauli Ift140* mutant [22]. The *Ift140$^{220}$* mutation was identified as a T to C transition in the splicing consensus sequence in intron 9 of *Ift140* (*IVS9+2T>C*), thus predicting it is hypomorphic with intron retention and premature termination with nonsense mediated decay. Consistent with this, the *Ift140$^{220/220}$* mutant mice survived to term. To assess for SBDs, necropsy was conducted, followed by serial histological imaging with episcopic confocal microscopy (ECM). ECM allowed detailed visualization of anatomical defects with rapid digital reslicing of the 2D histological image stacks and also rapid high-resolution 3D reconstructions [28].

Examination of younger embryos among the *Ift140$^{220/220}$* mutants showed cranial neural tube closure defects (Fig 1A and 1B) and left-right patterning defects with 19% of embryos exhibiting reversal of heart looping orientation (*n* = 27) (Fig 1C and 1D). Analysis of older embryos revealed a wide spectrum of other SBDs, including exencephaly, anophthalmia, craniofacial defects with bilateral facial cleft, cleft palate, and shortened snout with hypoplasia of the maxilla and mandible and shortened limbs with polydactyly (Fig 1F and 1G and Table 1). Also observed in the older embryos/fetuses were omphalocele, skin tags over the chest and abdomen that are reminiscent of supernumerary mammary glands (Fig 1H), hydroureter and kidney cysts, and duplex/multiplex kidneys (Fig 1H–1Q and Table 1). A spectrum of complex CHD is also observed (Fig 1K–1N and S1 and S2 Movies) including persistent truncus arteriosus (PTA) indicating failure in outflow tract (OFT) septation (S3 and S4 Movies), pulmonary artery stenosis, OFT malalignment defects with double outlet right ventricle (DORV) or overriding aorta (OA), and atrioventricular septal defects (AVSD) (Table 1). Additionally, hypoplasia of the aorta, aortic valve atresia, and coarctation of the aorta were observed.

**Table 1. SBDs associated with 2 different mouse *Ift140* mutant alleles.**

| Embryos harvested | Genotype | *n* | Phenotypes |
|---|---|---|---|
| E13.5-E17.5 | *Ift140$^{220/220}$* | 55 | Reversed heart looping, RAA, PTA, PA Stenosis, AVSD, Excencephaly (35%), Omphalocele (82%), Macrostomia (96%), Polydactly (100%), Skin Tags (60%), Eye Defects (58%), Hydrops (57%) |
| E9.5-E10.5 | *Ift140$^{null1/null1}$* | 22 | Randomized heart looping (D-looped 41%; L-looped 36%; A-looped 23%); NTD (95%) |
| E14.5 | *Ift140$^{null1/null1}$* | 7 | Randomized heart looping (D-looped 43%; L-looped 14%; A-looped 43%), PTA (100%), RAA (43%), Double AA (29%), AVSD (100%), exencephaly with complete failure in neural tube closure (100%), macrostomia (100%), hydrops (71%), TEF (100%), hypoplastic lungs (100%), diaphragmatic hernia (100%), polydactyly (100%). |

AA, aortic arch; AVSD, atrioventricular septal defect; NTD, neural tube closure defect; PA, pulmonary artery; PTA, persistent truncus arteriosus; RAA, right aortic arch; SBD, structural birth defect; TEF, tracheoesophageal fistula.

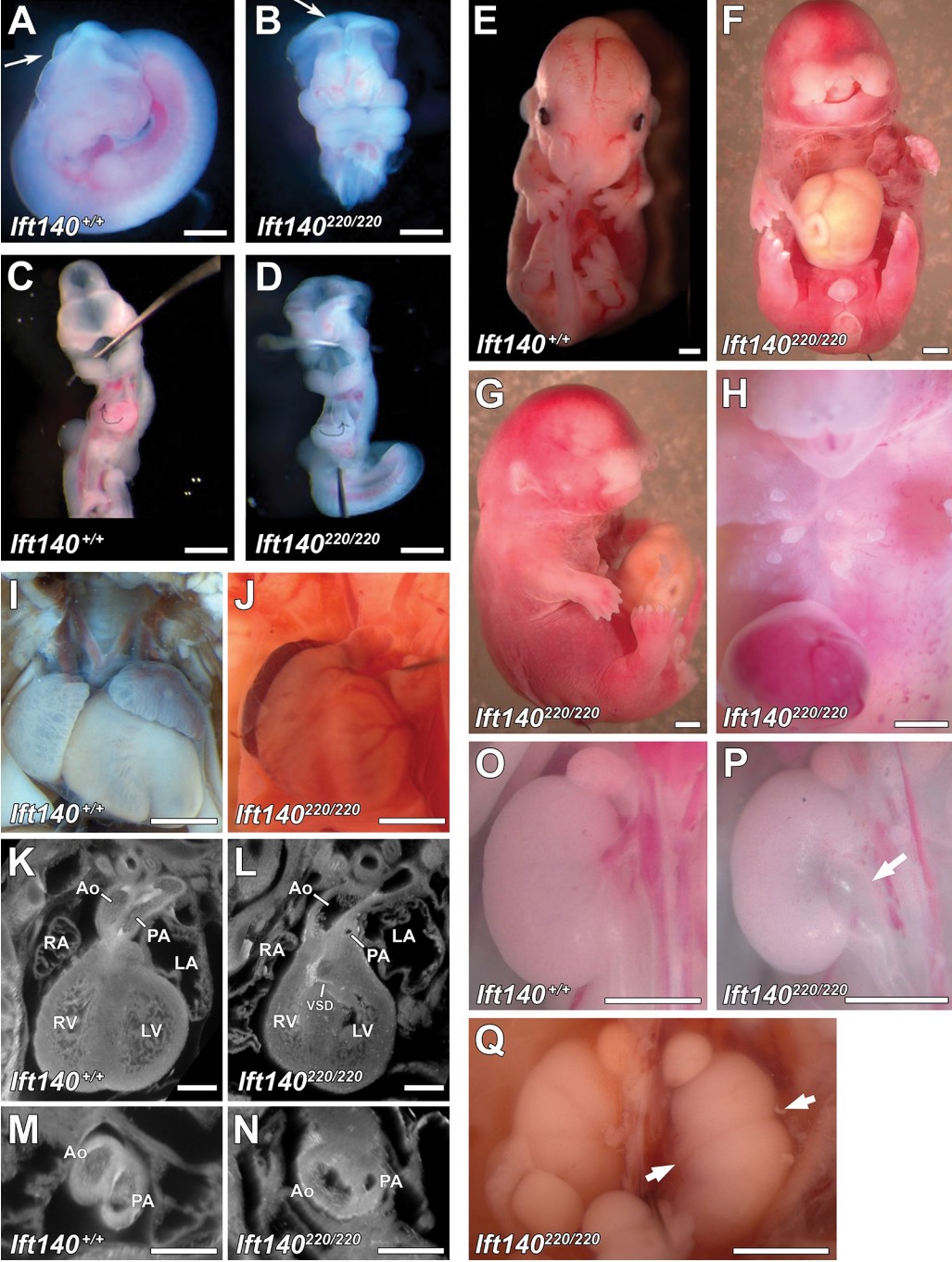

**Fig 1. *Ift140$^{220/220}$* mutant mouse embryos display a range of developmental defects.** (A–D) Gross anatomy of E11.5 embryos reveals *Ift140$^{220/220}$* mutants have open neural tube defect (arrows in A and B) and defective heart tube looping (arrows in C and D). (E–H) Gross anatomy of E14.5 embryos reveals *Ift140$^{220/220}$* mutants display a range of developmental defects including severe craniofacial defects (F, G), anophthalmia (G), omphalocele (F, G, H), polydactyly (G), and abnormal chest skin tags (H) that may represent abnormal mammary tissue development. (I, J) Cardiac anatomy of E16.5 embryos reveals *Ift140$^{220/220}$* mutants display VSDs (K, L) and abnormal OFT development (M, N) including PA stenosis and aorta (Ao) dilation (N). (O–Q) Renal anatomy of E14.5 embryos reveals *Ift140$^{220/220}$* mutants display hydroureter (P arrow) and duplex kidney (Q arrows highlight constrictions between duplex kidneys). Scale bars: **A–J, O–Q** = 1 mm, **K–N** = 0.5 mm. OFT, outflow tract; PA, pulmonary artery; VSD, ventricular septal defect.

## Midgestation lethality of *Ift140* knockout mice

To investigate impact of constitutive loss of *Ift140* function, an *Ift140* targeted ES cell line generated by KOMP was obtained and used to generate mice bearing knockout (KO) or null (*Ift140*$^{null1}$) alleles of *Ift140* [23]. Breeding of double heterozygous *Ift140*$^{null1}$ mice yielded the expected mendelian ratios until E14.5. At E13.5, 10 of the 15 mutant embryos obtained were dead and beyond E13.5, fewer mutants were observed than expected, with none surviving to term (S1 Table). For phenotyping of SBDs, efforts were made to collect the occasional mutant embryo not yet resorbed at E14.5. As a result, 7 E14.5 mutant embryos were obtained (Figs 2 and 3), and all 7 mutants exhibited severe SBD phenotypes (Table 1). This included hydrops, shortened limbs with polydactyly, exencephaly and severe craniofacial defects with hypoplasia of the upper face and tongue, facial clefting, and maldevelopment of the maxillary and mandibular prominences associated with wide a gaping mouth, clinically known as macrostomia (Fig 2A–2L). Omphalocele and ectopic cordis were observed with the liver/gut (omphalocele) and heart protruding outside the abdominal wall (Fig 2E and 2G). While omphaloceles are relatively common (1 in 5,000 live births) [29], ectopia cordis is very rare (approximately 1 in 5 to 8 million births) [29]. The diaphragm failed to form (Fig 2D and 2H), resulting in the liver projecting into the thoracic cavity, and lung development appeared to be developmentally arrested (Fig 2H and S7 and S8 Movies). Quantification of the chest showed a significant reduction in chest volume (Fig 2M), supporting the ciliopathy associated narrow trunk and thoracic dystrophy phenotypes. Also observed are brain malformations including microcephaly with severe forebrain hypoplasia. This can be observed in conjunction with exencephaly (Fig 2E–2G and 2K).

Examination of younger *Ift140*$^{null1/null1}$ embryos showed at E9.5, gross enlargement of the first branchial arch, while the remaining arches were hypoplastic (Fig 3A and 3B). Neural tube closure defects (NTD) were also observed, with some embryos showing complete failure of the head fold to elevate and fuse (Fig 3C–3F). Similar to the *Ift140*$^{220}$ mutant, heart looping was abnormal. While normal D-looped hearts were observed in all wild-type littermates, the *Ift140* null mutants exhibited either D-loop, reversed L-loop (Fig 3G–3M), or A-loop hearts in which the heart tube failed to loop to the right or left, but instead projected outward from the embryo (Figs 3I and S1 and Table 1). As changes in OFT length have been associated with OFT septation and malalignment defects, we also measured the OFT length. This showed significant OFT lengthening in the *Ift140* KO mouse heart (Fig 3N). Similar to the homozygous *Ift140*$^{220}$ mutants, the *Ift140*$^{null1}$ mutant embryos showed complex CHD (Fig 3O–3V), including PTA (Fig 3S and 3U) and AVSD (Fig 3T). Tracheoesophageal fistulas (TEFs) comprising fusion of the trachea and esophagus were also observed (Fig 3V and S5 and S6 Movies). Additionally, aortic arch defects were observed comprising right-sided aortic arch (RAA), or double aortic arch forming a vascular ring with the aorta connected to either a right- or left-sided aortic arch (Fig 3W).

## *Ift140* expression and ciliogenesis

To investigate impact of *Ift140* deficiency on ciliogenesis, mouse embryonic fibroblasts (MEFs) were generated from the homozygous *Ift140*$^{220}$ and *If140*$^{null1}$ embryos. Real-time PCR analysis of RNA from *Ift140*$^{220/220}$ MEFS showed a low level *Ift140* transcripts that are reduced to 15% of that seen in wild-type MEFs, thus confirming the *Ift140*$^{220}$ mutation is hypomorphic (Fig 4A). In contrast, the *Ift140*$^{null1}$ mutation reduced transcripts from remaining downstream exons to 2% of wild-type levels (Fig 4A). Western blotting with an antibody generated against the C-terminal end of mouse IFT140 did not detect any protein in either the *Ift140*$^{220/220}$ or *Ift140*$^{null1/null1}$ MEFs (Fig 4B). This is consistent with premature termination expected from

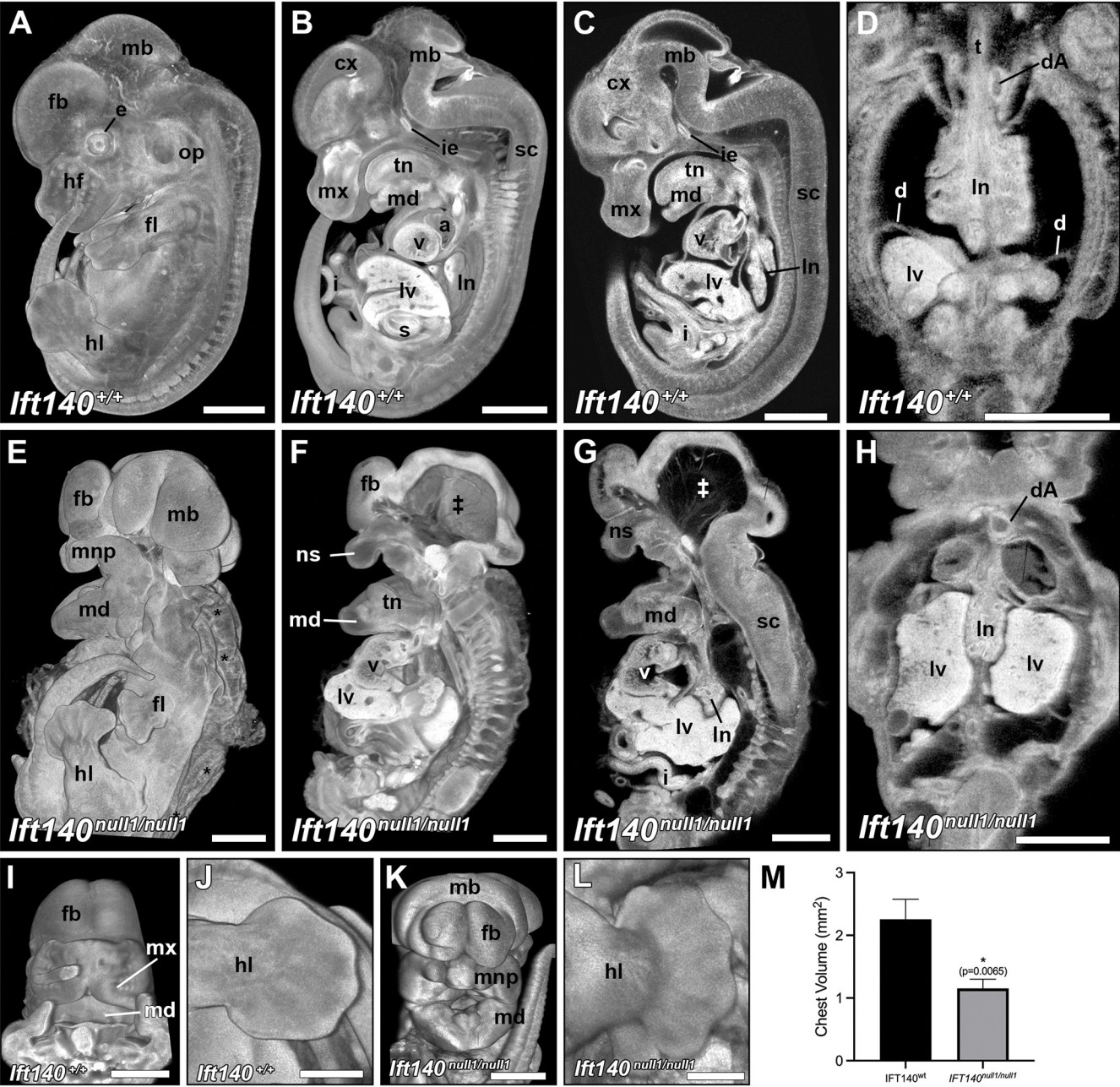

**Fig 2. *Ift140*$^{null1/null1}$ embryos display major anatomical defects at E14.5.** Gross anatomical examination revealed numerous severe defects in E14.5 *Ift140*$^{null1/null1}$ embryos (E–H) compared to controls (A–D) including significant hydrops (*), hypoplastic forelimbs (fl), hypoplastic maxillary region (mx), including reduced maxillary, medial and lateral nasal prominences resulting in bilateral cleft lip, hyperplastic mandibular region (md), missing abdominal walls and diaphragm with gastroschisis/ectopia cordis (F, G), smaller chests (D vs. H), and exencephaly with swollen neural tissue pouches surround an empty hollow cavity (‡). (I–L) 3D reconstitutions highlight the craniofacial defects (I, K) and polydactyly (J, L) found in E14.5 *Ift140*$^{null1/null1}$ embryos. (M) Chest size was quantified by measuring chest areas that revealed that *IFT140*$^{null1/null1}$ embryos (*n* = 7) displayed significantly smaller chests than age matched wild-type embryos (*n* = 3) (unpaired Students *t* test; *p* = 0.0065). cx: cerebral cortex; d: diaphragm; dA: descending aorta; e: eye; fb: forebrain; fl: forelimb; hl: hindlimb; hf: hair follicles; i: small intestine; ie: inner ear; ln: lung; lnp: lateral nasal prominences; lv: liver; mb: midbrain; md: mandibular region; mnp: medial nasal prominence; mx: maxillary region; ns: nasal capsule; op: otic placode; s: stomach; sc: spinal cord; t: trachea; tn: tongue; v: ventricle. Scale bars: A–I, K = 1 mm, J, L = 0.5 mm. The data underlying this figure can be found in Supporting information S1 Data.

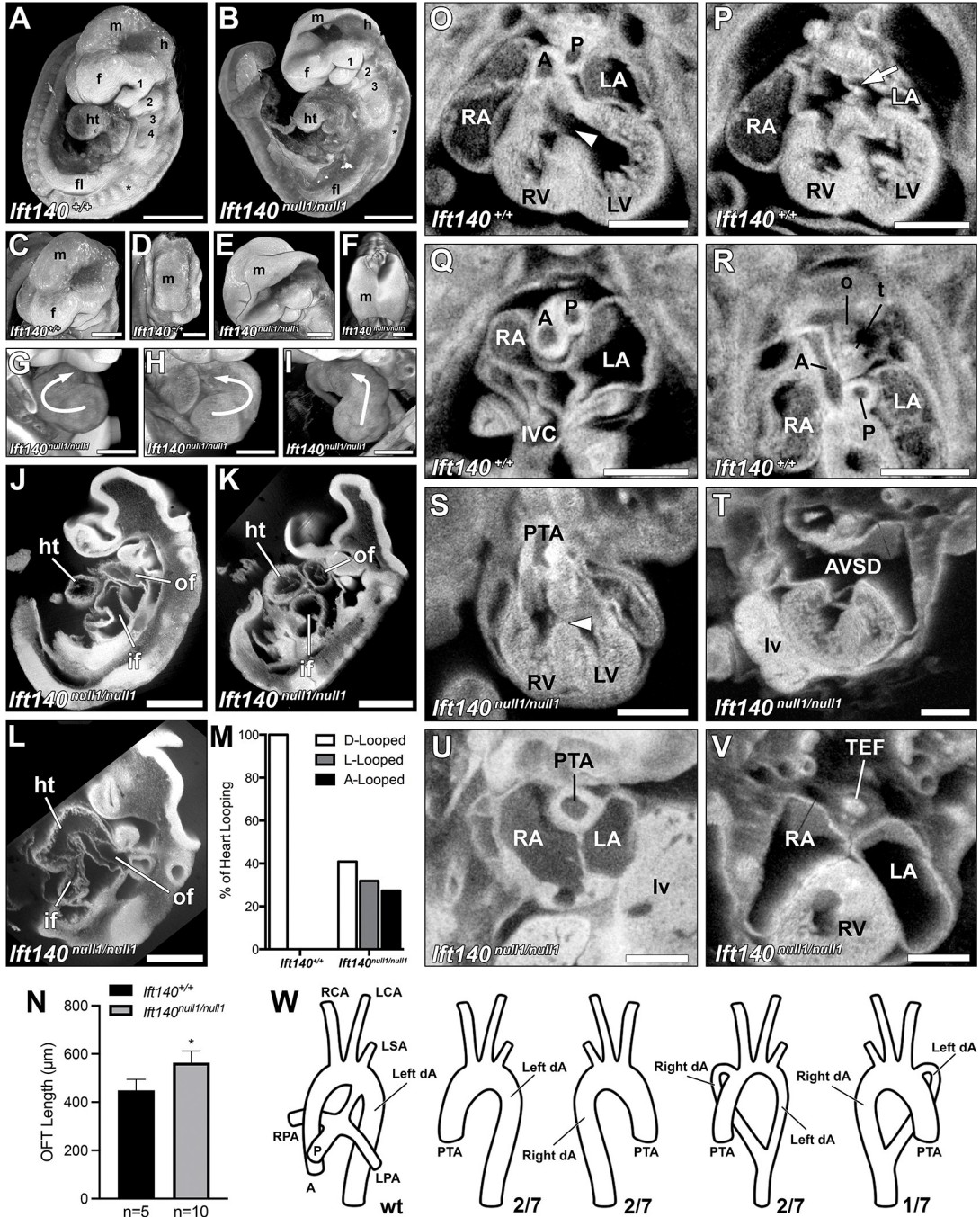

**Fig 3. *Ift140^{null1/null1}* embryos display major anatomical defects at E9.5 and cardiac/great vessel patterning defects at E14.5.**
(A–I) 3D reconstructions of E9.5 *Ift140^{+/+}* control and littermate homozygous *Ift140^{null1/null1}* embryos analyzed by episcopic confocal microscopy. In **A-B** note the hypertrophy of the first branchial arch (1), and hypotrophy of the other branchial arches (2–4). In **C-F** note neural tube abnormalities characterized by the head fold failing to close in *Ift140^{null1/null1}* embryos (**E-F**). In **G-I** note the randomization of heart tube looping characterized by normal D-looped (**G**), reversed L-looped (**H**), and midline A-looped (**I**) heart tubes. (J–L) Sagittal section reconstructions of *Ift140^{null1/null1}* embryos further highlight the abnormal midline A-looped heart tube phenotype (**L**) observed in some *Ift140^{null1/null1}* embryos compared to embryos with D-looped (**J**) or L-looped (**K**) heart tubes. (M) Quantification of the *Ift140^{null1/null1}* heart looping defects reveals the randomization of looping phenotypes compared with wild-type embryos (*n* = 22). (N) Measurement of the OFT length in E10.5 embryos showed significant lengthening of the OFT in the Ift14O KO embryos. Data is mean ± SD. * *p* = 0.0007 assessed by unpaired Student *t* test. (O–V) Numerous cardiac and great vessel defects were seen in E14.5 *Ift140^{null1/null1}* embryos including: small ventricles (**O** vs. **S**), AVSDs with mutant embryos displaying a complete absence of normal atrial septum (arrow **P** vs. **T**), PTA characterized

by a single OFT due to OFT failing to septate into separate aorta and pulmonary vessels (**Q** vs. **U**), and TEF characterized by a single unseptated tracheoesophageal tube (**R** vs. **V**). (W) Great vessel patterning was also perturbed in $Ift140^{null1/null1}$ embryos ($n$ = 7). Besides PTA, mutants showed a combination of singular or double left and right descending aortas. *: somites; 1, 2, 3, 4: Branchial arches; A: aorta; AVSD: atrioventricular septal defect; dA: descending aorta; f: forebrain; fl: forelimb; h: hindbrain; ht: heart tube; if: inflow tract; LA: left atrium; LCA: left carotid artery; LSA: left subclavian artery; LV: left ventricle; lv: liver; m: midbrain; o: esophagus; of: outflow tract; P: pulmonary trunk; PTA: persistent truncus arteriosus; RA: right atrium; RCA: right carotid artery; RV: right ventricle; t: trachea; tn: tongue; TEF: tracheoesophageal fistula; arrowhead: ventricular septal defect; arrow: atrial septum. Scales bars: **A, B, J, K, L, N–U** = 0.5 mm, **C–I** = 0.25 mm. The data underlying this figure can be found in supplemental file S1 Data.

anomalous transcripts arising from the $Ift140^{220}$ splicing defect mutation and no transcript expression from the $Ift140^{null1}$ allele (Fig 4A and 4B).

Analysis of ciliogenesis showed 13% of $Ift140^{220/220}$ and 2% of $Ift140^{null1/null1}$ MEFs were ciliated as compared to approximately 50% of wild type (Fig 4D). The cilia on $Ift140^{220/220}$ cells were stumpy and accumulated IFT B protein IFT88, while in $Ift140^{null1/null1}$ cells, IFT88 was found at one or more of the centrosomes (Fig 4C). MEFs from the $Ift140^{null1/null1}$ line did not stain with Ift140 antibodies, whereas the control cells showed strong staining at the ciliary base and some staining at the tip of the cilia (Fig 4C). Interestingly, even though MEFs from the $Ift140^{220/220}$ mutant showed no protein expression by western blotting, most ciliated cells showed Ift140 immunostaining at the base of the cilium or at 1 centrosome in non-ciliated cells, consistent with the low level of transcript expression detected (Fig 4C). This residual protein expression may account for the finding of more ciliated cells in the $Ift140^{220/220}$ versus $Ift140^{null1/null1}$ MEFs.

To investigate how Ift140 deficiency affected ciliogenesis in vivo, we examined primary cilia in the kidney of the $Ift140^{220/220}$ mutant embryos using immunofluorescence microscopy and observed abnormal bulbous accumulation of Ift88 at the distal tip of the cilia (Fig 4E). We also examined motile cilia in the embryonic node of E7.5-E7.75 embryos using scanning electron microscopy. As expected, most cells in wild-type nodes displayed a single 3 to 5 μm long cilium, but in the homozygous $Ift140^{220/220}$ mutant, most cilia in the embryonic node were shortened with a bulbous swelling (Fig 4F). In contrast, in the $Ift140^{null1/null1}$ mutant embryos, most cells in the embryonic node lacked cilia, except for an occasional cell with a short bulbous cilium (Fig 4F).

## Shh signaling defect associated with *Ift140* deficiency

IFT is known to regulate hedgehog signaling [2], a pathway important for many developmental processes. While IFT-B mutations typically inhibit hedgehog signaling, IFT-A mutations can either enhance or inhibit hedgehog signaling [30–32]. In fibroblast cells, the *Gli1* transcription factor is normally expressed at low levels, but expression is activated by stimulation of hedgehog signaling. Using the *Ift140* mutant MEFs, we assessed responsiveness to hedgehog stimulation with a hedgehog agonist by quantifying *Gli1* expression (Fig 5A). The $Ift140^{220}$ mutant MEF showed attenuated responsiveness to hedgehog stimulation, similar to findings in other *Ift* mutants [33], while $Ift140^{null1}$ mutant MEFs showed no *Gli1* up-regulation, indicating it is unresponsive to hedgehog stimulation (Fig 5A).

## *Ift140* deficiency perturbs developmental patterning of the neural tube

To investigate hedgehog signaling in vivo, we examined dorsoventral patterning of the neural tube in E10.5 $Ift140^{220/220}$ and $Ift140^{null1/null1}$ mutant embryos. This is a developmental process regulated by sonic hedgehog (Shh) expression in the floor plate and notochord [34] (Fig 5B). Immunostaining was used to examine expression of 3 dorsoventral neuronal differentiation

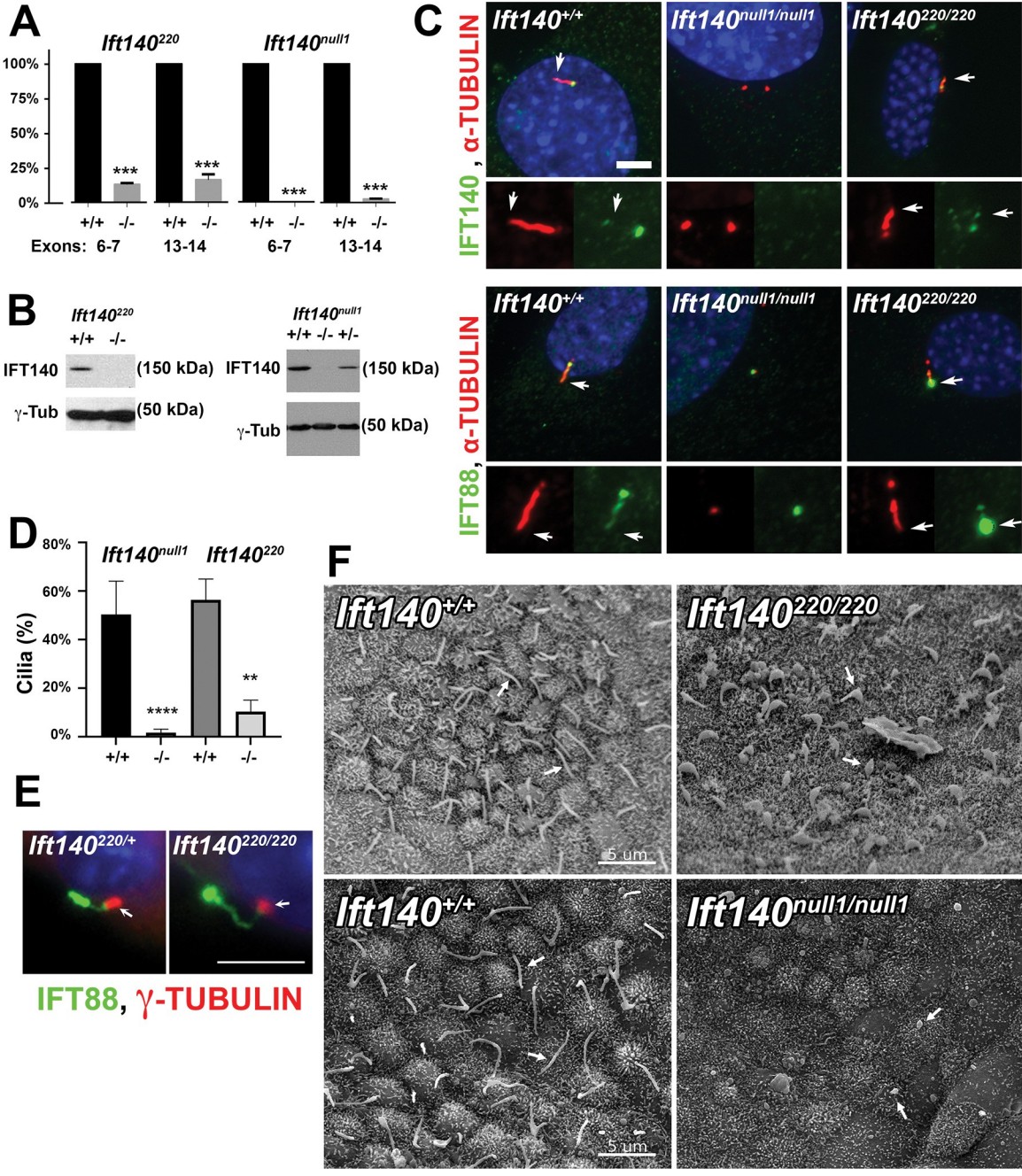

**Fig 4. Comparison of *Ift140²²⁰* allele and *Ift140^{null1}* allele.** (A) *Ift140* mRNA levels in MEFs. Levels of *Ift140* transcript between exons 6 and 7 and between exons 13 and 14 was measured by qPCR. Wild type was set to 100%. For *Ift140²²⁰/²²⁰*, $n = 1$ control cell line and 2 mutant lines repeated 3 times. For *Ift140^{null1/null1}*, $n = 3$ control lines and 3 mutant lines analyzed once each. *** $p \leq 0.001$, Students $t$ test. (B) Western blot analysis of IFT140 protein levels in MEFs from the 2 alleles. (C) MEFs from control and mutant lines stained for cilia (acetylated tubulin, red) and either IFT140 or IFT88 (green). 0% of *Ift140^{null1/null1}* cells showed IFT140 staining at the ciliary base or centrosome. $63 \pm 16\%$ of *Ift140²²⁰/²²⁰* cells showed weak IFT140 staining at the ciliary base while control cells for each experiment showed 100% of the ciliated cells showing an IFT140 spot at the ciliary base. For *Ift140²²⁰/²²⁰*, $n = 1$ control cell line and 2 mutant lines repeated 3 times. For *Ift140^{null1/null1}*, $n = 3$ control lines and 3 mutant lines analyzed once each. Scale bar is 5 microns. Arrows mark ciliary tip. (D) Percent ciliation in control and mutant fibroblast lines. For *Ift140^{null1}*, $n = 100$ cells counted from 3 repeats of 1 control line and 1 mutant line. ***** $p < 0.0001$ $t$ test. For *Ift140²²⁰*, $n = 100$ cells counted from 3 repeats of 1 control and 2 *Ift140²²⁰/²²⁰* mutant lines. ** $p = 0.0037$ $t$ test. (E) Cilia on Bowman's capsule of the kidney stained for centrosomes (γ-tubulin, red, arrow) and IFT88 (green). Scale bar is 5 microns. (F) Scanning EM images of control, *Ift140²²⁰/²²⁰*, and *Ift140^{null1/null1}* embryo nodes harvested at E7.5. Arrows indicate cilia. Scale bar is 5 microns. The data underlying this figure can be found in Supporting information S1 Data. MEF, mouse embryonic fibroblast.

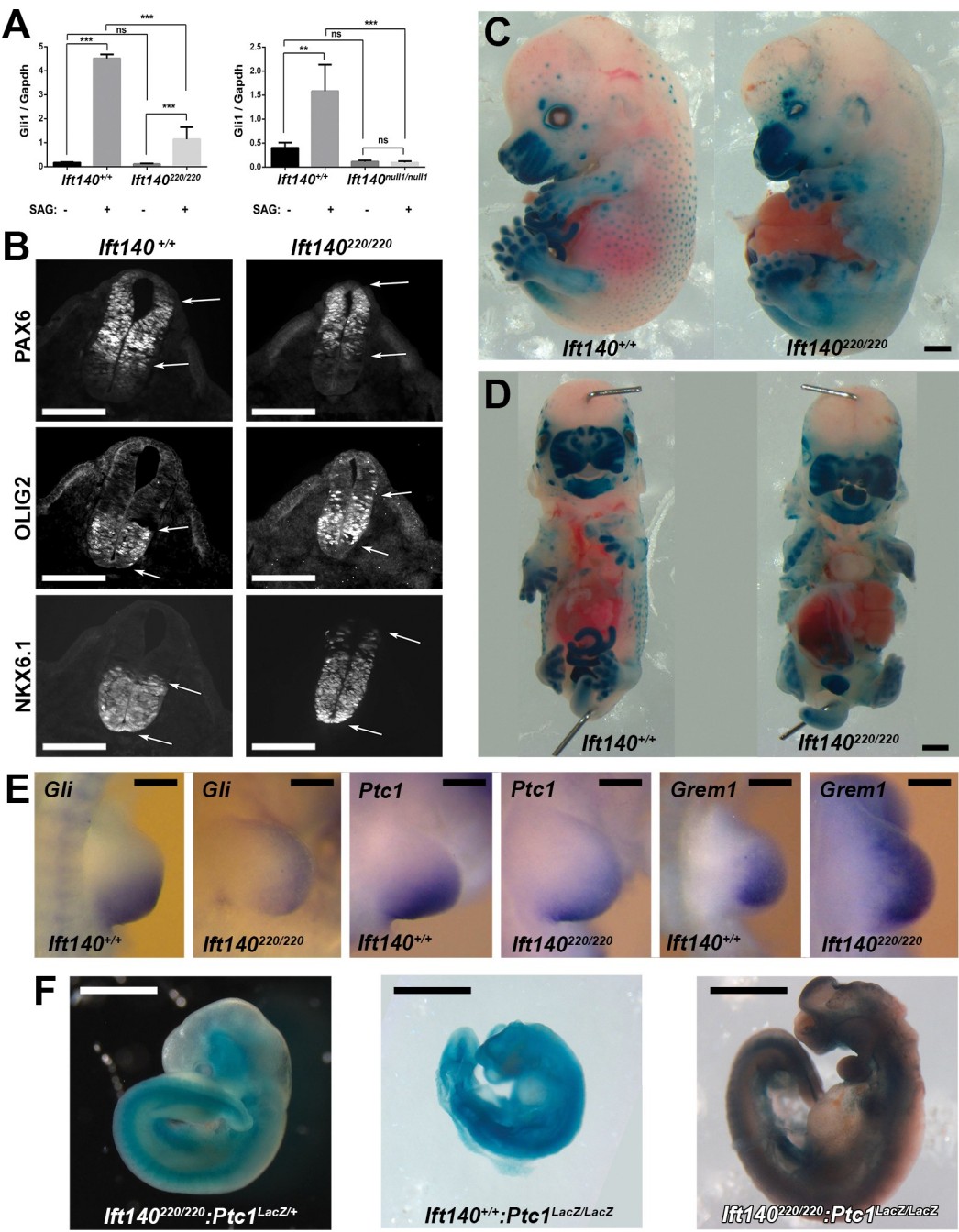

**Fig 5. Perturbation of hedgehog signaling associated with developmental defects in the _Ift140^{null1/null1}_ and _Ift140^{220/220}_ mutant embryos.** (A) To assess hedgehog signaling in cultured MEFs, RNA was isolated from cells that were left untreated or treated with 400 nM SAG for 24 h. _Gli1_ and _Gapdh_ gene expression was measured by quantitative real-time PCR. For _Ift140^{220}_ cells, $n = 1$ control and 2 mutant lines analyzed 3 times. For _Ift140^{null1}_ cells, $n = 3$ control and 3 mutant lines each analyzed 1 time. $**p \leq 0.01$; $***p \leq 0.001$; ns, not significant, assessed by one-way ANOVA. (B) Immunostaining for differentiation markers in the neural tube of E10.5 embryos revealed disturbance in _Shh_ regulated dorsoventral patterning of the neural tube in the _Ift140^{220/220}_ mutants. Ventralization is indicated with dorsal shift in expression of ventral markers OLIG2, and NKX6.1, and dorsal retraction in expression of PAX6, a dorsal marker. Arrows denote the boundaries of antibody staining. (C, D) Sagittal and Frontal views of E14.5 wild-type and _Ift140^{220/220}_ mutant embryos carrying a _Gli-LacZ_ reporter, delineating regions of hedgehog signaling. (E) In situ hybridization of limb buds from E10.5 wild-type and _Ift140^{220/220}_ mutant embryos showed perturbation of _Shh_ signaling, with expanded expression of Gremlin indicating polydactyly. (F) _Ift140^{220/220}_ (left), _Ptch1^{LacZ/LacZ}_ (middle), and _Ift140^{220/220}:Ptc1^{LacZ/LacZ}_ (right) E10.5 embryos carrying the Ptc1-LacZ knockout allele were X-gal stained to delineate regions of hedgehog signaling. The severe phenotype of the

*Ptc1$^{LacZ/LacZ}$* mutant embryo is partially rescued in the double homozygous *Ift140$^{220/220}$:Ptc1$^{LacZ/LacZ}$* mutant embryo. Scales bars: **B** = 100 μm, **C, D, F** = 1 mm, **E** = 0.25 mm. The data underlying this figure can be found in supplemental file S1 Data. MEF, mouse embryonic fibroblast.

markers including PAX6, a dorsal marker, and NKX6.1 and OLIG2, 2 ventral markers. *Ift140$^{220/220}$* mutant embryos displayed a dorsal shift in expression of ventral markers Olig2, and Nkx6.1, and a dorsal retraction in expression of Pax6, a dorsal marker (Fig 5B), indicating ventralization of neural tube patterning from increased hedgehog signaling.

NKX6.1 normally expressed in the ventral half of the neural tube with demarcation of a discrete boundary, showed no change in the ventral domain of expression in the *Ift140$^{220/220}$* mutants (Fig 5B). However, individual NKX6.1-positive cells were observed ectopically above the boundary demarcating the dorsal half of the neural tube, indicating ventralization of cell fates (Fig 5B). Ventralization was also indicated with analysis of OLIG2 expression, which is normally found in a discrete band starting just above the floor plate and extending dorsally to the middle of the neural tube. In *Ift140$^{220/220}$* mutants, OLIG2 expression expanded dorsally, with isolated cells seen even in the vicinity of the roof plate (Fig 5B). Together, these findings suggest increased hedgehog signaling in dorsoventral patterning of the neural tube in *Ift140* mutant embryos.

## Shh signaling and structural birth defects in the *Ift140* mutant mice

To further investigate Shh disturbance in the SBD phenotypes seen the *Ift140* mutant mice, a *Gli1* lacZ insertion KO allele referred to as Gli1-LacZ [35] was intercrossed into the *Ift140$^{220}$* mouse line. Homozygous *Ift140$^{220}$* mutant mice heterozygous for the Gli-lacZ allele were harvested at E14.5, and whole mount X-gal staining was conducted to visualize regions of Shh activity (Fig 5C and 5D). In the wild-type littermate, strong lacZ expression was observed in the upper (maxilla) and lower jaw (mandible), and in the fore and hindlimbs, all regions affected by significant SBDs in the *Ift140* mutants. Strong X-gal staining was preserved in these regions in the *Ift140$^{220}$* mutants, but the distribution showed subtle changes reflecting the anatomical alterations associated with the SBDs (Fig 5C and 5D). While elevated X-gal staining was noted in the tongue, this may reflect better substrate perfusion from the facial clefts and foreshortened snout (Fig 5D). Expression of *Gli1* and *Ptch1* in the limb buds was examined by in situ hybridization as both genes are downstream targets of Shh signaling. Both genes were down-regulated even as their domains of expression expanded anteriorly (Fig 5E). Conversely, expression of *Gremlin* was elevated and expanded anteriorly consistent with the polydactyly phenotype (Fig 5E).

A *Ptch1* LacZ insertion KO allele (*Ptch1-LacZ*) [36] was also intercrossed into the *Ift140$^{220}$* mutant mice to further assess impact of the *Ift140* mutation on hedgehog signaling (Fig 5F). Homozygous *Ptch1-LacZ* mice exhibited early postimplantation lethality associated with developmental arrest at E8.5 [36]. Mice double homozygous for both the *Ift140$^{220}$* and *Ptch1-LacZ* mutations showed rescue of the *Ptch1-LacZ* lethality, with double homozygous mutant embryos surviving to E10.5 (Fig 5F). This suggests reduced Shh signaling in the *Ift140$^{220/220}$* mutant, indicating *Ift140* deficiency may have provided partial recovery from the *Ptch1-LacZ* KO allele gain-of-function effects on Shh signaling.

## Temporal requirement for *Ift140* in left-right patterning

Mice carrying the tamoxifen (Tmx)-inducible *CAGGCre-ER* were intercrossed with the floxed *Ift140* (*Ift140$^{flox}$*) allele to investigate temporal requirement for *Ift140* in left-right patterning. Previous studies showed *CAGGCre-ER* mediated deletion can be observed 10 h after tamoxifen

treatment [37]. Using western blotting, we found that embryos collected 48 h after tamoxifen treatment had only 19% of the Ift140 protein remaining (S2 Fig). Hence, Tmx treatment was conducted at least 24 h before the developmental process to be targeted. Given inherent developmental asynchrony, Cre deletion within a litter may vary by 0.5 day or more. Mice homozygous for *Ift140* floxed allele were adult viable with no phenotype. Mice carrying the Cre alone also had no phenotype except for some simple muscular ventricular septal defects (VSDs), phenotype also seen in some wild-type embryos (Table 2).

To investigate role of *Ift140* in left-right patterning, Tmx treatment was conducted between E5.5 to 7.5, as nodal cilia regulating left-right patterning are present between E7.5 to E8.0 [7]. With Tmx treatment at E7.5, embryos collected at E12.5/14.5 showed no heart looping defects (Table 2), indicating *Ift140* is required before ~E8.0–8.5 for left-right patterning. Earlier Tmx treatment at E5.5 and 6.5 caused developmental arrest with embryonic lethality, necessitating earlier embryo collection at E11.5 to 12.5 (Fig 6). Tmx treatment at E5.5 yielded 14 Cre +/*Ift140* floxed embryos, 9 (64%) had D-loop, 3 (21%) L-loop, and 2 (14%) A-loop heart (Fig 6). Tmx treatment at E6.5 yielded 1 embryo with D-loop, and 2 with A-loop heart. Overall, this replicated the heart looping defects observed in the *Ift140^{null1}* and *Ift140^{220}* mutant embryos, indicating early requirement for cilia in left-right patterning.

**Table 2. SBD phenotypes associated with Cre targeted *Ift140* deletion.**

| Cre driver[†] | Tamoxifen administered | Embryos harvested | Genotype[††] | n | Phenotypes |
|---|---|---|---|---|---|
| *CAGGCre-ER* | E5.5, 6.5 | E11.5-E12.5 | Control | 7 | Normal |
| | | | Cre+ | 17 | Heart looping defects* (D-looped 59%; L-looped 18%; A-looped 23%), AVSD (18%), developmental delay (35%) macrostomia (12%), NTD (6%) |
| *CAGGCre-ER* | E7.5 | E12.5-E14.5 | Control[†] | 2 | Normal (mVSD) |
| | | | Cre+ | 16 | PTA (19%), polydactyly (81%), preductal CoA (12%), IAA (6%), MAPCA (12%), hypoplastic lungs (19%), hydrops (19%), cleft palate (25%), macrostomia (25%), omphalocele (100% at E14.5), ectopia cordis (100% at E14.5) |
| *CAGGCre-ER*[†] | E7.5, E8.5 | E16.5 | Control | 3 | Normal (mVSD) |
| | | | Cre+ | 11 | Dilated thin RV (73%), DORV (9%), OA (9%), IAA (27%), polydactyly (100%), hydrops (82%), multiple VSDs (73%), ASD (9%), macrostomia (100%), omphalocele (80%), ectopia cordis (80%) |
| *Foxa2Cre-ER*[†] | E6.5-E8.5 | E14.5-E16.5 | Control | 4 | normal (mVSD, 25%) |
| | | | Cre+ | 8 | normal (mVSD, 25%) |
| *Wnt1-Cre*[†] | NA | E14.5-E18.5 | Control | 4 | normal |
| | | | Cre+ | 5 | normal heart looping, IAA (40%), AA with abnormal Ao branching (60%), cleft palate (100%), macrostomia (100%), omphalocele (100%) |
| *Mef2c-Cre*[†] | NA | E16.5-E18.5 | Control | 1 | normal (mVSD) |
| | | | Cre+ | 8 | normal (mVSD, 63%) |
| *Tbx18-Cre*[†] | NA | E16.5 | Control | 2 | normal |
| | | | Cre+ | 9 | polydactyly (56%), pVSD (33%), PV stenosis (11%)hydrops (89%), cleft palate (33%), macrostomia (11%), abdominal skin tags (89%), omphalocele (22%), ectopia cordis (11%) |
| *Tie2-Cre*[†] | NA | E18.5-E20.5 | Control | 0 | - |
| | | | Cre+ | 8 | Normal (mVSD, 63%), abdominal skin tags (25%), eyelid defect (25%), omphalocele (12%) |

[†]*CAGGCre-ER* and *Foxa2Cre-ER* expression driven by tamoxifen gavage at stages listed. All other Cre drivers expressed via respective promoters.
[††]Littermate controls with genotypes: Cre- or Cre+ with *Ift14^{null1/flox}*, *Ift140^{flox/flox}*, or *Ift140^{+/+}*.
*Of 17 embryos, 3 were from E6.5 Tmx treatment, 1 had D loop (33%), 2 had A (66%) loop. Of 15 embryos from E5.5 Tmx treatment, 9 (64%) had D loop (21%), 3 (21%) had L looped, and 2 (14%) had A loop.
AA, aortic arch; Ao, aorta/aortic; CoA, coarctation; IAA, interrupted aortic arch; MAPCA, multiple aortopulmonary collateral arteries; mVSD, muscular VSD; NTD, neural tube closure defects; OA, overriding aorta; PTA, persistent truncus arteriosus; pVSD, perimembranous VSD; PV, pulmonary valve; RV, right ventricle; SBD, structural birth defect.

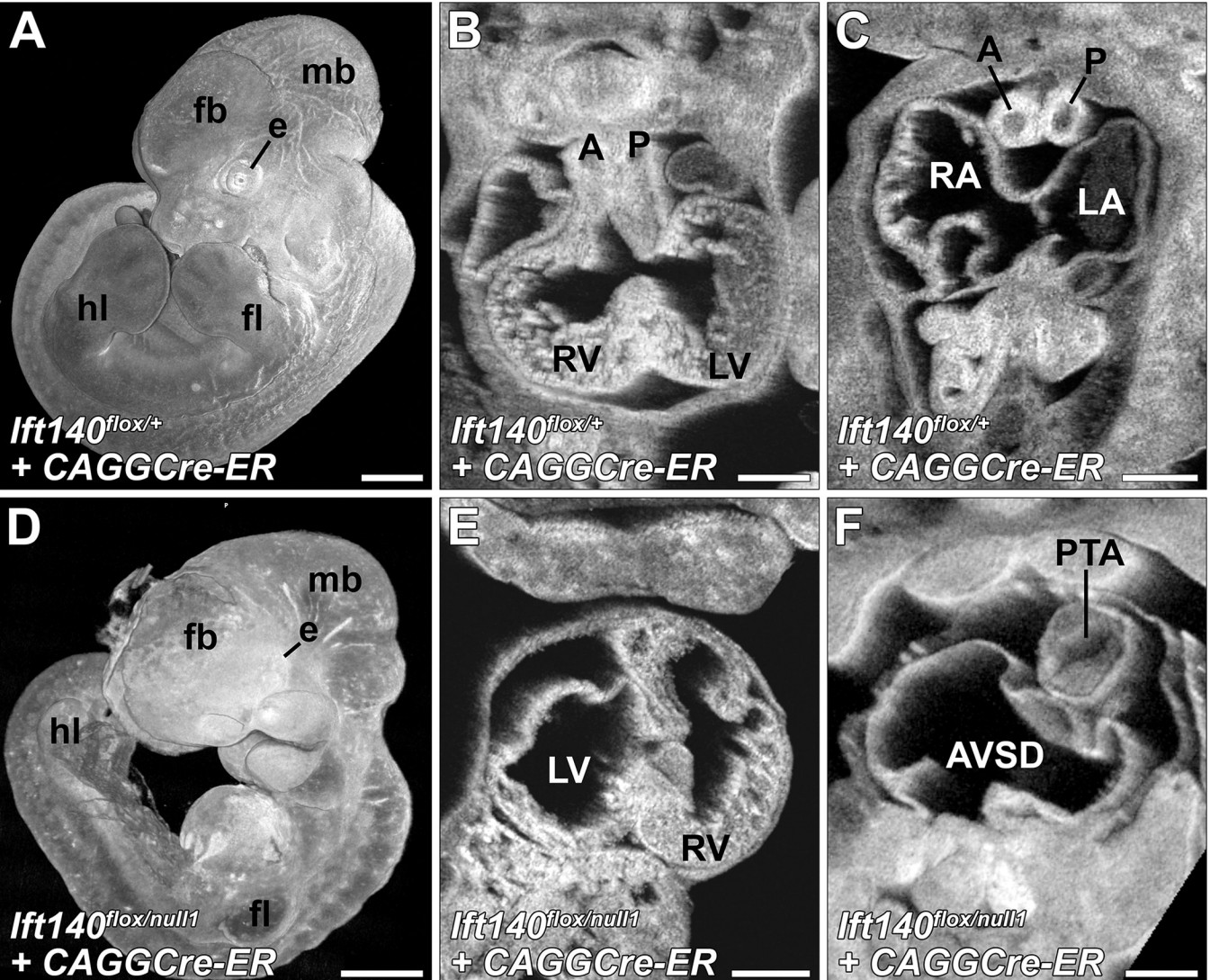

**Fig 6. Early tamoxifen deletion of *Ift140* with *CAGGCre-ER* recapitulates the *Ift140*[null1/null1] phenotype.** *Ift140*[flox/+], *CAGGCre-ER*[+] (Control) (**A–C**) and *Ift140*[flox/null1], *CAGGCre-ER*[+] experimental (**D–F**) embryos were treated with tamoxifen at E5.5 and harvested at E12.5. The experimental embryos had extensive developmental abnormalities and showed developmental delay (**A, D**). The experimentals recapitulated the laterality defects seen in *Ift140*[null1/null1] embryos as characterized by reversed heart tube looping and the morphological right ventricle appearing on the embryo's left side (**B** vs. **E**). Similar to the *Ift140*[null1/null1] embryos, tamoxifen-driven deletion of *Ift140* using *CAGGCre-ER* at E5.5 also caused atrial septal defects and outflow track septation defects (PTA) (**C** vs. **F**). However, head fold closure defects and exencephaly were not observed under these conditions. fb: forebrain; mb: midbrain; fl: forelimb; hl: hindlimb; e: eye; A: aorta; P: pulmonary trunk; LV: left ventricle; RV: right ventricle; LA: left atria; RA: right atria; PTA: persistent truncus arteriosus. Scales bars: **A, D** = 0.5 mm, **B, C, E, F** = 0.25 mm.

The temporal requirement for *Ift140* in left-right patterning was further investigated using Tmx inducible *Foxa2Cre-ER*, which specifies expression in the embryonic node [37]. Tmx treatment at E6.5, E7.5, or E8.5 did not result in laterality defects (Table 2 and S3 Fig). As expression of *Foxa2Cre-ER* driven LacZ expression is reported to be patchy in the node of E7.5 embryos, this would suggest inadequacy of this Cre driver for gene deletion in the embryonic node [38]. While *FoxA2* is also expressed in the endoderm, notochord, and floor plate, no other SBD phenotypes were observed, negative findings that may also be impacted by the patchy expression of the Cre driver.

## Temporal requirement for *Ift140* in the structural birth defects phenotypes

The temporal requirement for *Ift140* in the broad spectrum of SBDs seen in the *Ift140* KO and *Ift140*[220] mutant mice were further interrogated with Cre deletions conducted at E7.5 or 8.5, and embryos collected at E11.5–12.5, E12.5–14.5, or at E16.5 (Table 2). No heart looping defects were observed in any embryos generated (Table 2 and Fig 7), consistent with the earlier requirement for cilia in left-right patterning. While analysis of the older stages allowed more complete SBD phenotyping, this might have caused bias towards milder phenotypes, since more severely affected embryos may not survive to the later stages (Figs 6 and 7 and Table 2). Consistent with this, embryos collected at E12.5–14.5 from Tmx treatment at E7.5 had more severe range of phenotypes than those collected at E16.5, suggesting a subset of embryos were likely dying and resorbed before harvest at E16.5. Hence, E16.5 embryos obtained from Tmx treatment at E7.5 and E8.5 were grouped together, comprising the milder group, while E12.5–14.5 embryos from Tmx treatment at E7.5 comprised the severe group. Collectively, most of the SBD phenotypes seen in the *Ift140*[null1] mice were replicated, including aortic arch defects, macrostomia, omphalocele and ectopia cordis, and polydactyly (Fig 7). However, the lung hypoplasia was observed only in the severe E7.5 group.

In the severe group, CHD was observed comprising persistent truncus arteriosus (PTA) indicating complete failure in OFT septation (Table 2 and Fig 6). This CHD phenotype was not observed in the mild group (Table 2 and Fig 7), which exhibited only OFT malalignment defects, such as DORV or overriding aorta (OA), VSDs and atrial septal defects (ASDs) (Table 2 and Fig 7J–7M). While aortic arch anomalies were observed in both the mild and severe Tmx treatment groups (Fig 7N), major aortopulmonary collateral arteries (MAPCAs) were observed only in the severe group. These findings suggest development of the OFT and aortic arch vessels require *Ift140* function surprisingly early, perhaps at E8.5–9.5 (24 h after Tmx treatment), which is several days before formation of the four-chamber heart or aortic arch arteries.

## Lineage-specific requirement for *Ift140* in the SBD phenotypes

To investigate cell lineage-specific requirement for *Ift140* in orchestrating the different SBD phenotypes observed in the *Ift140* mutant mice, different Cre drivers in combination with the floxed *Ift140* allele was used to target *Ift140* deletion in different lineages (Table 2). Focusing on interrogating the developmental etiology of the cardiovascular defects, we conducted Cre deletion using *Mef2c-Cre* to target the anterior heart field [39], *Tie2-Cre* for endothelial/endocardial cells [40], *Wnt1-Cre* for neural crest [41,42], and *Tbx18-Cre* for epicardial cells [43]. *Mef2c-Cre* (*Ift140*[+/flox]: *Mef2c-Cre*) deletion yielded only small VSDs (S3 and S4 Figs and Table 2). As small VSDs were also observed in control mice treated with Tmx, these may reflect nonspecific Tmx treatment effects. Postnatal follow up of the four *Mef2C-Cre* deleted *Ift140*[-/flox] mice showed they are adult viable with no obvious phenotypes. For *Tie2-Cre* deletion, no phenotypes were observed in the E18.5 mice, except for eye lid defects (S3 Fig), and skin tags that were also observed in the homozygous *Ift140*[220] mutant mice.

For *Tbx18-Cre* (Figs 8 and 9), expression is expected in the pharyngeal area, proepicardium and epicardium [44], somites, limb bud and progenitor of the vibrissae [43]. *Tbx18-Cre Ift140* deletion caused hydrops (Fig 8H) and polydactyly (Fig 8M) with high penetrance, but cardiovascular anatomy (Fig 9D–9F) was largely unaffected except for some perimembranous VSDs (Fig 9E and Table 2). Craniofacial structures also were unaffected except 1 in 10 *Tbx18-Cre* deleted *Ift140* embryos exhibited severe craniofacial defects with macrostomia and hyperplasia of the craniofacial prominences (Fig 8I). Body wall closure defects comprising omphalocele and ectopia cordis were observed (Table 2). Also observed were polyp-like skin tags on the

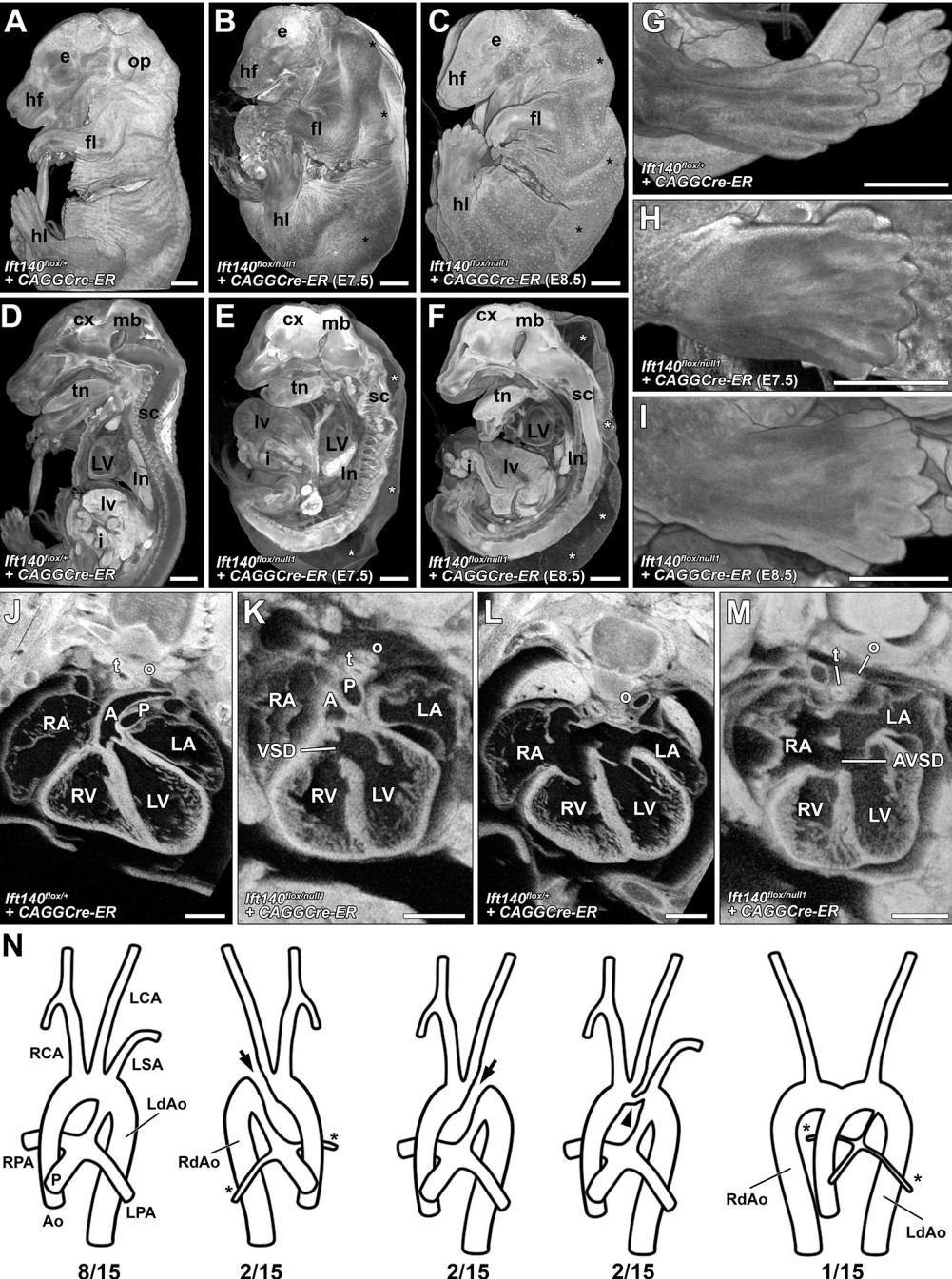

**Fig 7. Late tamoxifen deletion of *Ift140* with *CAGGCre-ER* uncovers additional phenotypes.** *Ift140^{flox/+}*, *CAGGCre-ER^+* (Control) (**A, D, G, J, L**) and *Ift140^{flox/null1}*, *CAGGCre-ER^+* experimental (**B, C, E, F, G, H, K, M, N**) embryos were treated with tamoxifen at E7.5 or E8.5 and harvested at E16.5. E7.5 tamoxifen-dosed embryos show severe gastroschisis with the majority of the abdominal organs protruding from the abdominal cavity (**A, B, D, E**). E8.5 dosed embryos show only moderate gastroschisis with only some of the abdominal organs found outside of the abdominal cavity (**A, C, D, F**). Both E7.5 and E8.5 dosed embryos showed significant hydrops (\* **B, C, E, F**), polydactyl (**H, I**), and hypoplastic lungs (**B, C, E, F**). Laterality defects were not observed in either E7.5 or E8.5 tamoxifen-dosed embryos, with all hearts displaying a normal D-looping phenotype (**J, K**). However, cardiac defects were observed in the experimental animals including ventricular septal defects (**J, K**) and AVSDs (**L, M**). (N) While the great vessels of both E7.5 and E8.5 tamoxifen-dosed experimental embryos displayed normally septated aorta and pulmonary trunk, approximately 50% had great artery patterning defects including: right aortic arch, interrupted aorta (arrows), hypoplastic transverse aorta (arrowhead), hypoplastic pulmonary arteries (\*), and in 1 case double aortic arch with both left and right descending aortas. A: aorta; P: pulmonary trunk; LV: left ventricle; RV: right ventricle; LA: left atria; RA: right atria; t: trachea; o:

esophagus; VSD: Ventricular septal defect; AVSD: atrioventricular septal defect; cx: cerebral cortex; sc: spinal cord; mb: midbrain; fl: forelimb; hl: hindlimb; e: eye; forebrain; op: otic placode; hf: hair follicles; RCA: right carotid artery; LCA: left carotid artery; LSA: left subclavian artery; LdAo: left descending aorta; RdAo: right descending aorta; RPA: right pulmonary artery; LPA: left pulmonary artery; Ao: aorta; P: pulmonary trunk; lv: liver; ln: lungs; s: stomach; i: small intestine. Scales bars: A–I = 1 mm, J–M = 0. 5 mm.

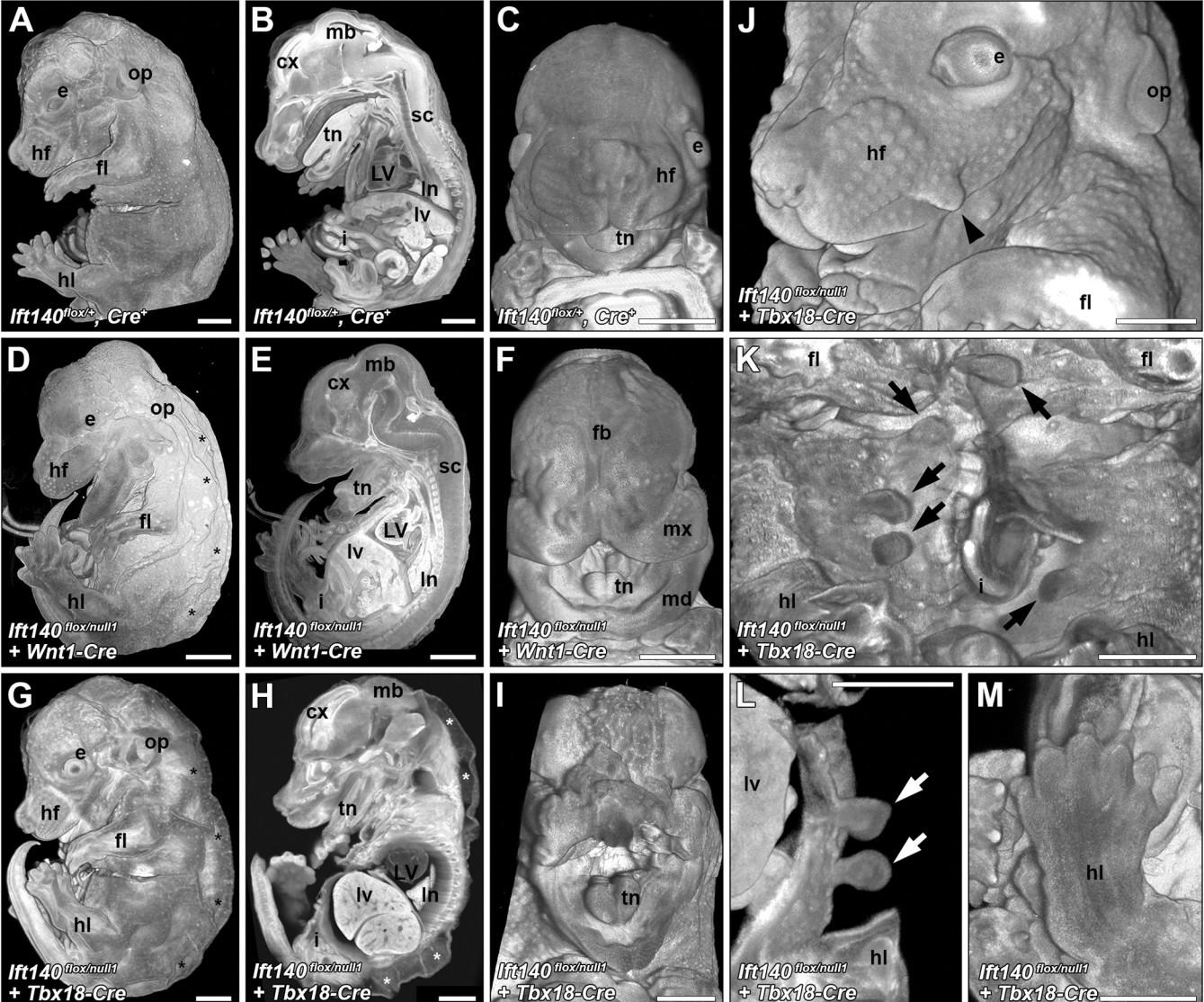

**Fig 8. Gross anatomical defects in embryos with *Wnt1-Cre* or *Tbx18-Cre* deletion of *Ift140*.** (A–C) Littermate control embryos (*Ift140^flox/+*, *Wnt1-Cre⁺* or *Tbx18-Cre⁺*) displayed normal embryonic anatomy. (D–F) *Wnt1-Cre* deletion of *Ift140* resulted in a 100% penetrative phenotype characterized by significant hydrops (asterisks in **D**) and marked craniofacial defects including macrostomia and hypertrophied forebrain, maxillary, and mandibular regions (**F**). (G–M) *Tbx18-Cre* deletion of *Ift140* resulted in severe hydrops (asterisks in **G, H**), but less severe cranial facial defects (**H**). While most embryos *Tbx18-Cre* experimental embryos displayed normal craniofacial anatomy (**G, H**), a single embryo (1/10) displayed a wide mouth phenotype reminiscent of a bird's beak as well as marked cranial tissue hypertrophy (**I**). Embryos with *Tbx18-Cre* deletion of *Ift140* displayed a number of skin protrusions located to the face (arrowhead in **J**) and more commonly to the abdomen (arrows in **K, L**). Polydactyly was also observed in *Tbx18-Cre* experimental embryos (**M**). LV: left ventricle; cx: cerebral cortex; sc: spinal cord; fb: forebrain; mb: midbrain; fl: forelimb; hl: hindlimb; e: eye; op: otic placode; hf: hair follicles; lv: liver; ln: lungs; s: stomach; i: small intestine; mx: maxillary region; md: mandibular region. All scales bars = 1 mm.

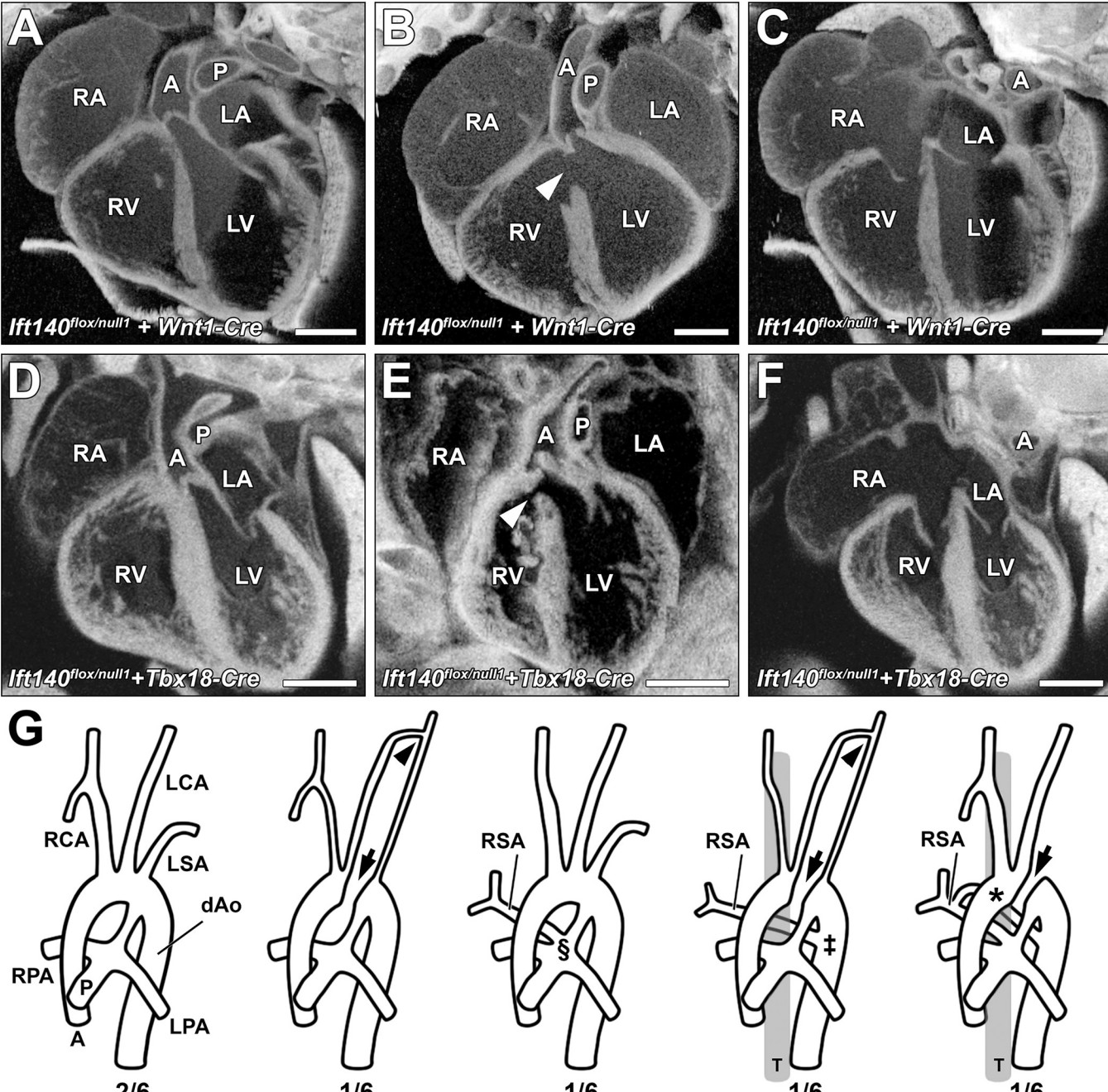

**Fig 9. Cardiac and great vessel defects with *Wnt1-Cre* or *Tbx18-Cre* deletion of *Ift140*.** (A–F) Both *Wnt1-Cre* and *Tbx18-Cre* driven *Ift140* deletion (*Ift140^{flox/null1}*, *Cre^+*) produced only mild congenital cardiac defects. All experimental hearts displayed normal heart looping and the majority had normal ventricular (**A, D**) and atrial (**C, F**) septum anatomy. However, a small number of *Wnt1-Cre* and *Tbx18-Cre* experimental animals displayed perimembranous ventricular septal defects (arrowheads in **B, E**). (G) *Wnt1-Cre* driven *Ift140* deletion caused defects in great artery patterning including interrupted aorta (arrows in **G**), with or without the development of a long hypoplastic collateral vessel linking the left carotid artery and left subclavian artery (arrowheads in **G**). The development of anomalous right subclavian arteries was common in *Wnt1-Cre* experimental embryos (*Ift140^{flox/null1}*, *Wnt1-Cre*). These arose from either the pulmonary trunk adjacent to the pulmonary arteries (§), as a vascular sling arising from the descending aorta and wrapping behind the trachea (‡), or as a vascular ring with attachments to both the pulmonary trunk and descending aorta (*). A: aorta; P: pulmonary trunk; LV: left ventricle; RV: right ventricle; LA: left atria; RA: right atria; T: trachea; RCA: right carotid artery; LCA: left carotid artery; LSA: left subclavian artery; RSA: right subclavian artery; dAo: descending aorta; RPA: right pulmonary artery; LPA: left pulmonary artery; T: trachea. All scales bars = 0.5 mm.

ventrolateral abdomen similar to those seen with *Tie2-Cre* deletion (Fig 8K and 8L). Skin tags were also observed on the face (Fig 8J) and appeared to be associated with one of 2 prominent hair follicles in the snout (see plate 49 in [45]). Measurement of the chest volume showed no significant difference from control littermates.

Using *Wnt1-Cre Ift140* deletion, we investigated the role of *Ift140* in the neural crest lineage. Subpopulations of neural crest cells comprising the cardiac and cranial neural crest play critical roles in cardiac and craniofacial development. *Wnt1-Cre* deletion caused omphaloceles and craniofacial defects with complete penetrance (Table 2 and Fig 10). The latter included macrostomia with malformed maxillary and mandibular prominences and cleft palate (Fig 8A–8F). The enlarged maxilla and abnormally shaped mandibular process were evident by E12.5 (yellow lines, Fig 10C). This was associated with abnormal positioning of the eyes behind the enlarged maxilla (Fig 10C). Examination of the craniofacial defects with skeletal preps of E18.5 fetuses stained with Alizarin red and Alcian blue (Fig 10) showed fusions between the upper and lower jaw resulting in the loss of the temporomandibular joint (Fig 10G). In the cranial vault, ectopic bony islands (arrowhead in Fig 10H) were observed near the frontal bones reminiscent of cell migration defects seen with craniosynostosis, a phenotype also seen in the Gli3Xt-j mutant and craniosyntosis phenotypes are common in many syndromic ciliopathies affecting IFT [46]. The mandible was malformed, being shorter and broader, and missing the 3 processes (Fig 10K and 10L).

Surprisingly, cardiac anatomy was largely unaffected by *Wnt1-Cre* deletion of *Ift140* (Fig 9A–9C), except for some perimembranous VSDs (Fig 9B). Interrupted aortic arch and collateral vessels were observed, such as ectopic vessels connecting the left carotid with the left subclavian arteries (Fig 9G). Also observed were MAPCAs, such as anomalous right subclavian arteries arising from the pulmonary trunk, vascular sling formed by collateral vessel from the descending aorta encircling the trachea, or vascular ring comprised of collateral vessel emerging from the pulmonary trunk extending to the descending aorta (Fig 9G). Together, these findings indicate *Ift140* deficiency in neural crest cells contribute to the aortic arch vessel abnormalities, and craniofacial defects, but surprisingly, *Ift140* deletion in neural crest cells did not replicate the cardiac OFT septation and malalignment defects observed in the *Ift140*[220] or *Ift140* KO mice.

## Discussion

A wide spectrum of SBDs was observed in mice harboring a splicing or null allele of *Ift140* encoding an IFT-A component required for ciliogenesis. In contrast to mid-gestation lethality of the *Ift140*[null1] allele, the *Ift140*[220] splicing mutation is viable to term, suggesting it is hypomorphic. This is supported by detection of a low level of *Ift140* transcripts in the homozygous *Ift140*[220] mutant MEFs. Both *Ift140* mutants exhibited a similar wide array of SBD, including left-right patterning defects with randomization of heart looping, macrostomia, exencephaly and neural tube closure defects, polydactyly, body wall closure defects with omphalocele and ectopia cordis, diaphragmatic hernia, CHD with OFT septation and malalignment defects, and AVSD. Lung development appeared developmentally arrested and is associated with a very small chest reminiscent of the thoracic dystrophy seen in SRTD. This lung underdevelopment could be related to the lack of a diaphragm and the marked expansion of liver tissue into the thoracic cavity of these embryos, which may physically prevent normal lung growth and expansion. In some mutants, TEFs were also observed. It should be noted that some SBD phenotypes were identified only by examining the older fetuses surviving from the *Ift140*[220] mutant line. This included cleft palate, vascular rings/slings, renal anomalies comprising multiplex kidneys, kidney cysts, hydroureter, and skin tags. Skin tags found on the abdomen are

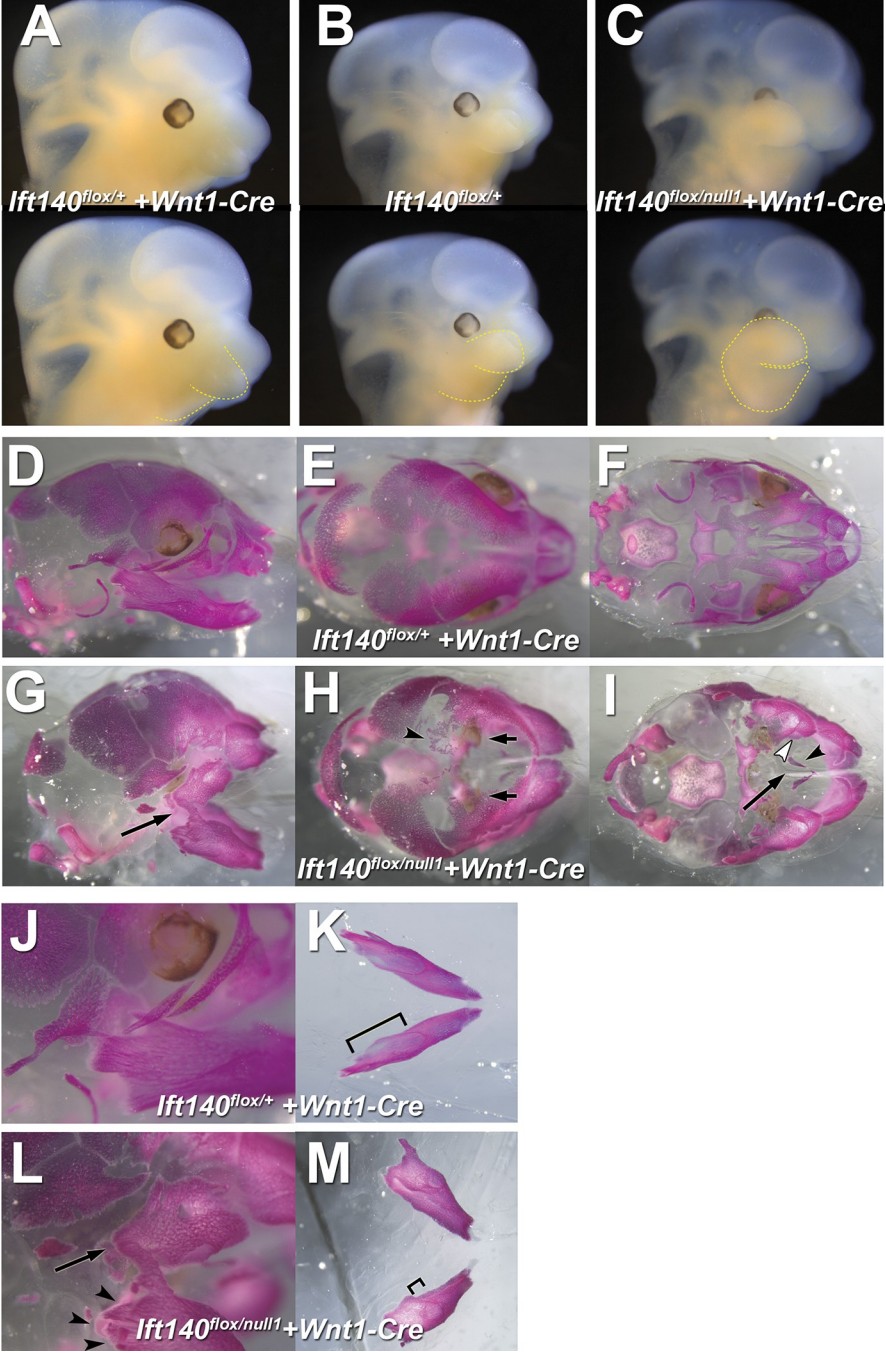

**Fig 10. Hyperplasia of maxillary and mandibular prominences resulting in bone fusions with *Wnt1-Cre* deletion of *Ift-140*.** (A–C) E12.5 control (*Ift140^flox/+^*, *Wnt1-Cre^+^* or *Wnt1-Cre^-^*) (**A, B**) and experimental animals (*Ift140^flox/null1^*, *Wnt1-Cre^+^*) (**C**). Overgrowth of the maxillary and mandibular processes (yellow outline, bottom row) is apparent in the mutant animals (**C**). The maxillary overgrowth is concealing the eye. (D–O) E18.5 control (**D–F, J–L**) and experimental animals (**G–I, M–O**). The *Wnt1-Cre Ift140^flox/null1^* mutant skull and face is shortened, and smaller (**G–I**) than the *Ift140^flox/+^ Wnt1-Cre^+^* control embryos (**D–F**), and have several defects in neural crest-derived bones. Laterally, the temporomandibular joint is absent resulting in the fusion between the maxilla and mandible in the experiment animals (**G, L** arrows). In the bird's eye view of the skull, there are ectopic boney islands present in the mutant frontal bones (arrowhead, **H**), suggestive of a problem with cell migration. Remarkably, the eyes are visible in this view but are below the frontal bones and medial to the maxilla (**H,** arrows). The palatal view shows that the maxillary bones are displaced laterally (**I,** open arrowhead), the vomer is present (**I,** arrowhead) and the anterior, neural crest-derived cranial base is absent (**I,** arrows). (J–O) The mandible is missing its 3 processes (arrowhead, **L**). The alveolar ridge for the molars is present, but is smaller (arrowhead, **M**).

reminiscent of those previously reported in the Gli3 mutant mice and may be malformed supernumerary mammary glands [47].

That these SBD phenotypes are related to defects in ciliogenesis is supported by in vitro analysis of *Ift140* mutant MEFs that showed a low incidence of ciliation and in vivo analysis showing prominent cilia defects in the kidney and in the embryonic node of the *Ift140* mutant mice. Moreover, defects in cilia transduced Shh signaling was indicated by the in vitro analysis of MEFs. This is supported by additional in vivo analyses that showed disturbance of anterior-posterior limb patterning associated with the polydactyly, ventralization of the neural tube, and partial rescue of the *Ptch-LacZ* early embryonic lethality by *Ift140*[220]. Clinically, *Ift140* mutations are associated with classic skeletal ciliopathies known as SRTD [11–17], with patients exhibiting skeletal abnormalities with a small thoracic cavity. Additionally, craniosynostosis may be observed with fusion of bones in the head and jaw that are cranial neural crest/cranial mesenchyme derived. Individuals also can have early onset retinal degeneration and cystic kidney disease [12–14]. Many of these phenotypes are also observed in the *Ift140* mutant mice. While some of the severe SBDs observed in the mouse model are not reported in patients, such defects might compromise postnatal survival and thus biasing surviving patients to only those harboring the milder hypomorphic alleles. Consistent with this, *IFT140* ciliopathy patients often are compound heterozygous [48,49], with 1 allele being null and a second allele with a missense mutation. We note *Ift140* null mice die at E11-13, equivalent to human gestation days 30 to 44 [45,50]. This would predict embryonic death in the first trimester.

The randomized heart looping observed is consistent with the known requirement for motile and primary cilia in the embryonic node for left-right patterning [51]. The temporal Cre deletion experiments indicated cilia requirement is prior to E8.5, as heart looping defects were observed only with Tmx treatment at E5.5/6.5, but not at E7.5 (Fig 11). Furthermore, reversal of heart looping was only observed with Tmx treatment at E5.5, but not at E6.5. While this is prior to node formation, it is in line with the time required for the gene products to be degraded or diluted by cell division after the *Ift140* deletion. Aside from left-right patterning defects, most of the SBD phenotypes observed in the *Ift140* mutants were replicated by the temporal Cre deletion at E7.5/8.5, including craniofacial defects with macrostomia, body wall closure defects, and polydactyly. Aortic arch anomalies, OFT septation failure (PTA) and MAPCAs (E7.5) were observed only with Cre deletion at E7.5, while somewhat later Cre deletion (8.5) yielded milder CHD phenotypes comprising OFT malalignment defects (DORV/OA). Lung hypoplasia was observed with Tmx treatment at E7.5, but not at E8.5, suggesting *Ift140* is required at E8.5 or earlier for proper lung development (Fig 11).

Further analysis using the lineage specific Cre drivers yielded unexpected insights and complexities into the role of cilia in the developmental etiology of CHD and a wide spectrum of birth defects observed with *Ift140* deficiency. The 4 Cre drivers targeted the second heart field (SHF) that forms most of the OFT, the neural crest cells essential for OFT septation, endothelial/endocardial cells required for valvular morphogenesis, and the epicardium critical for coronary vascular development and ventricular septum development. While the CRE lines used in this study are well established and extensively utilized, the possibility exists that cardiac phenotypes were masked by variability in recombination levels and incomplete knockdown of ciliation in target tissues. Which would explain why, unexpectedly, none of these Cre drivers resulted in either cardiac outflow septation or malalignment defects, or the AVSD phenotypes seen in the *Ift140* mutant/KO mice. However, this has not been reported previously, and this is in sharp contrast to similar analysis for other genes, such as *Mef2C-Cre* deletion of *Lrp1* that fully replicated the DORV and AVSD phenotypes seen in the *Lrp1* KO mice [52].

While we found *Ift140*-deficient MEFs showed loss of cilia and cilia transduced Shh signaling, no CHD phenotype was observed with *Ift140* deletion using either the *Wnt1* or *Mef2c-Cre*

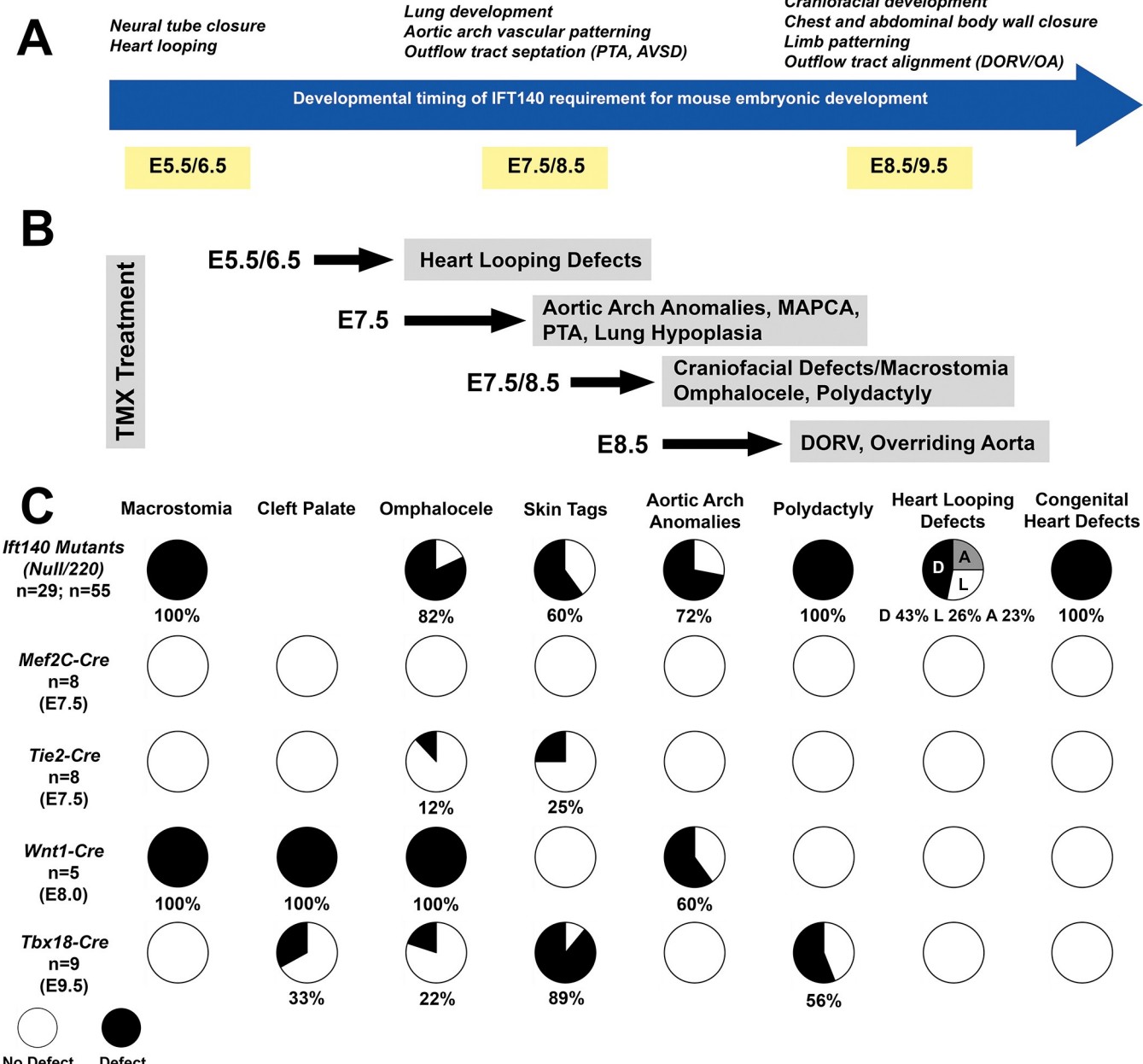

**Fig 11. Summary of the spatial and temporal roles for *Ift140* during embryonic development.** (A) The timeline shows the temporal requirement for *Ift140* revealed by tamoxifen mediated knock-out of *Ift140* at different time points during development. (B) Overview of embryonic days Tmx was used to knock down *Ift140* and the resultant phenotypes. (C) Diagram summarizing the cell lineage requirement for *Ift140* revealed by Cre-driven targeted *Ift140* knock-out. *Mef2c-Cre*: targeting cells in the anterior heart field; *Tie2-Cre*: targeting endothelial cells; *Wnt1-Cre*: targeting dorsal neural tube and neural crest; *Tbx18-Cre*: targeting epicardial cells, left ventricle.

driver. This is despite previous studies showing PTA with *Smo* deletion by either *Wnt1* or *Mef2c-Cre* and AVSD with *Mef2c-Cre* [53,54]. These findings would suggest the CHD phenotypes in the *Ift140* mutants are not likely to be the result of Shh deficiency. Consistent with this, we observed OFT lengthening in the *Ift140* mutants, while Smo or Shh deletion with *Mef2C-Cre* resulted in the opposite, the shortening of the OFT [53,54]. Shh has been shown to promote OFT lengthening by suppressing SHF differentiation into cardiomyocytes and

promoting SHF progenitor expansion, but with deficiency in Shh signaling, SHF progenitors are observed to undergo premature differentiation, resulting in OFT shortening. Thus, the CHD phenotypes in the *Ift140* mutants may not reflect Shh deficiency, but rather disturbance of other cell signaling pathways known to regulate OFT development (Fgf, Tgfb, Bmp, semaphorin, retinoic acid, Pdgf; [55]). However, given the additional finding of ventralization of the neural tube, another formal possibility is an increase in hedgehog signaling arising from the *Ift140* deficiency. This is reminiscent of observations in other IFT mutants—*Ift122^{sopb}* and *Ift139/Ttc21b^{aln}* mutants [31,32] and could account for OFT lengthening. Barring possible issues using Cre deleted models (where IFT140 may have been hypomorphic instead of null), the precise role of *Ift140* in hedgehog signaling may be context dependent, with OFT development mediated by a non-cell autonomous role of cilia in the integration of multiple cilia transduced cell signaling pathways across different cell lineages. This could provide functional redundancy to prevent the emergence of lethal CHD phenotypes. We note in addition to Shh, Tgfβ and Pdgf, are 2 other cilia transduced cell signaling pathways with essential roles in OFT septation and alignment [55].

In contrast to the CHD phenotypes, many noncardiac SBDs appeared to be cell autonomous and were replicated by the lineage-specific Cre drivers. Thus, *Wnt1-Cre* and *Tbx18-Cre* both yielded craniofacial defects (cleft palate, macrostomia) and omphaloceles. Replication of the craniofacial defects with the *Wnt1-Cre* is consistent with the known role of cranial neural crest contributing to head mesenchyme forming the maxilla and mandible, and other skeletal elements in the head and face. That this was replicated by *Tbx18* may seem surprising, but in fact *Tbx18* is also expressed in the head mesenchyme and segmentally in the rostral sclerotome, a compartment through which trunk neural crest cells migrate [56]. This segmental migration of neural crest cells reflect both migratory and avoidance cues [57]. The observation that both the craniofacial and body wall closure defects were replicated by *Tbx18-Cre* and *Wnt1-Cre* would suggest these phenotypes arise from the perturbation of the cranial and trunk neural crest, respectively, and may be cell autonomous. We note trunk neural crest cells migrating through the rostral scleratome compartment targeted by the Tbx18-Cre have been shown to give rise to skeletal muscle, with those migrating to form the ventral dermomyotome differentiating into hypaxial skeletal muscle of the diaphragm, abdominal wall, and limb [58]. Hence, it is likely that the loss of such trunk neural crest derived hypaxial musculature underlies the body wall closure defect observed with Tbx18-Cre deletion of *Ift140*. As *Tie2-Cre* also yielded omphalocele, this suggests blood vessels that are segmentally aligned with the sclerotome may also provide migratory cues to direct migration of neural crest cells to the ventral dermomyotome forming hypaxial muscles required for ventral body wall closure. As the rostral sclerotome will give rise to the ribs, whether trunk neural crest perturbation may contribute to thoracic dystrophy with shortened ribs warrants further investigation in future studies. Our finding that *Tbx18* replicated the polydactyly phenotype reflects the abundant expression of Tbx18 in the limb buds [43]. The prevalence of digit abnormalities in mice with cilia defects is complicated with various digit defects found in different alleles and within the same allele (e.g., Ift27 nulls show a variety of defects) [59]. It is likely that this reflects alterations in hedgehog signaling intersecting with less characterized pathways such as Wnt. Interestingly, the *Tie2-Cre* deletion also generated abdominal skin tags reminiscent of supernumerary mammary glands, and the eye lid defects, suggesting involvement of endothelial cells or blood vessel abnormalities in these defects.

Overall, our findings showed a broad spectrum of SBD phenotypes are elicited by *Ift140* deficiency (Fig 11). That the SBDs associated with CHD are orchestrated by non-cell autonomous events suggests unexpected complexity in heart development involving cilia orchestrated interactions between multiple lineages. In contrast, the extracardiac craniofacial and body wall

closure defects are mediated by cell autonomous role of cilia in the cranial and trunk neural crest, respectively. Interestingly, *Tbx18* is expressed in both the limb buds and in the rostral sclerotome that gives rise to the ribs. While *Tbx18-Cre* deletion of *Ift140* yielded polydactyly, whether this also may lead to short ribs will need to be further investigated with examination of older embryos. Given coincidental expression of *Tbx18* in the limb bud and rostral sclerotome, it is interesting to consider whether there is a functional link between *Tbx18* and short rib polydactyly (SRP). Our findings highlight the important role of cilia in a wide spectrum of SBDs and suggest the genetic risk associated with skeletal ciliopathies may involve a broader spectrum of SBDs that can include CHD.

## Materials and methods

### Study approval

Mouse research was carried out at the University of Massachusetts Chan Medical School with IACUC approval (PROTO201900265) and at the University of Pittsburgh with IACUC approval (Protocol 21120410). These IACUCs follow the regulations of US Department of Agriculture Animal Welfare Act and the standards/principles of the Public Health Service Policy on Humane Care and Use of Laboratory Animals, AVMA Guidelines on Euthanasia, US Government Principles for the Utilization and Care of Vertebrate Animals Used in Testing, Research and Training, and the Guide for the Care and Use of Laboratory Animals.

### Mouse breeding

The $Ift140^{220}$ mutant line was recovered from a large-scale mouse ENU mouse mutagenesis screen as previously described [26]. The $Ift140^{null1}$ and $Ift140^{flox}$ alleles were generated from ES cells with an *Ift140* targeted allele generated by KOMP [60] as described [23] (see S5 Fig for schematic of *Ift140* alleles used). Animals were genotyped with the following primers: 140komp2 TCAGCCCTCTATGCCACTCT, 140komp3 CTTCCCTATGCCTTCAGCAG, and 140komp6 TGGTTTGTCCAAACTCATCAA. The expected products sizes are 140komp3 +140komp2: WT = 190 bp, null1 = 0, flox = 269 bp, and 140komp2+140komp6: WT = 0, null1 = 328 bp, flox = 0.

Cre lines used include *CAGGCre-ER* [61], *Wnt1-Cre* [41,42], *Tbx18-Cre* [43], *Foxa2Cre-ER* [37], *Mef2c-Cre* [39,62], and *Tie2-Cre* [40] (S2 Table). Experimental animals were Cre+ and were $Ift140^{null1/flox}$ at the *Ift140* locus. Control animals were a mix of Cre+, $Ift140^{+/flox}$ and Cre-, $Ift140^{null1/flox}$ to control for any potential phenotypes arising from Cre expression or *Ift140* haploinsufficiency. Age of embryos was determined by timed mating with noon on the day of plug being E0.5. Tamoxifen (Sigma, St. Louis, Missouri, United States of America) was dissolved in vegetable oil at a concentration of 10 mg/ml and 0.1 ml (1 mg) was administered by oral gavage at noon (summary of mouse breeding used to generate embryos analysed in this study outlined in S3 Table).

LacZ reporter lines used include Gli1-LacZ ($Gli1^{tm2Alj}$/J; Jackson Laboratory Strain #:008211) [35] and Ptch1-LacZ ($Ptch1^{tm1Mps}$/J; Jackson Laboratory Strain #:003081) [36]. $Ptch1^{tm1Mps}$/J was intercrossed with the $Ift140^{220}$ mutant line to generate double heterozygous mice. These were further intercrossed to generate embryos for SBD phenotyping that are double homozygous for the $Ift140^{220}$ and $Ptch1^{tm1MPS}$ alleles.

### Structural birth defects phenotyping with episcopic confocal microscopy

Fetuses or neonatal pups were euthanized, fixed in 10% buffered formalin, paraffin embedded, and serially sectioned using a Lecia SM2500 microtome. Images of tissue autofluorescence

were collected from the block face during sectioning using a Lecia LSI confocal scan head, 488 nm excitation laser, and GFP emission filter settings (approximately 500 to 700 nm). Serial image stacks of the confocal images from the block face were collected. For 3D volume rendering, image z-stacks were generated using ImageJ [63] and processed using OsiriX (v.4.0 64-bit) [64]. To refine diagnosis of complex structural congenital defects, the 2D image stacks were viewed in different imaging planes by digitally resectioning in different imaging planes using OsiriX [28,65].

## Scanning electron microscopy

Embryos were fixed overnight in 2.5% glutaraldehyde in 0.1M sodium cacodylate. Fixed embryos were rinsed twice with 0.1M sodium cacodylate, osmicated in 1% osmium tetroxide, dehydrated in a graded ethanol series and critical point dried (Autosamdri-815, Series A, Tousimis Research Corp.). Dried embryos were sputter coated with iridium to a thickness of 3 nm (Cressington 208 HR Sputter Coater, Ted Pella, Redding, California, USA) and examined in a scanning electron microscope (FEI Quanta 200 FEG SEM) [66].

## Alizarin red staining for analysis of skeletal malformations

Fetuses were treated with 0.5% potassium hydroxide overnight and then stained in 0.1% alizarin red for 3 days. The embryos were subsequently further cleared with 1% KOH, 20% glycerol, and stored in 50% glycerol. Photography was performed in 50% glycerol on a Leica M165FC microscope.

## Immunohistochemistry

Embryos at E10.5 were fixed for 3 h in 4% paraformaldehyde in PBS, embedded in paraffin, and sectioned. Sections were deparaffinized and antigen retrieved by autoclaving for 40 min at 250°C. Primary antibodies OLIG2, PAX6, and NKX6.1 (Developmental Studies Hybridoma Bank, Univ. of Iowa) were detected with AlexaFluor labeled secondary antibodies (Life Technologies, Waltham, Massachusetts, USA).

## Analysis of mouse embryonic fibroblasts

MEFs were generated from E12 embryos by 30 min digestion with 1 ml of 0.25% trypsin/2.21 mM EDTA and early passage cells were immortalized with SV40 Large T antigen before analysis. MEFs were cultured in 90% DMEM (4.5 g/L glucose), 10% fetal bovine serum, 100 U/ml penicillin, and 100 μg/ml streptomycin (all from Gibco-Invitrogen, Carlsbad, California, USA). For SAG experiments, MEFs were plated at near confluent densities and serum starved (same culture medium described above but with 0.25% FBS) for 48 h prior to treatment to allow ciliation. SAG (Calbiochem, Billerica, Massachusetts, USA) was used at 400 nM and cells were treated for 24 h.

Cells for immunofluorescence microscopy were grown, fixed, and stained as described [33]. Primary antibodies used included acetylated tubulin (6-11B-1, Sigma), MmIFT27 [33], MmIFT88 [67], and MmIFT140 [23].

## Real-time PCR quantification of mRNA transcripts

To determine the amount of *Ift140* transcript that remained in the mutants, mRNA was isolated from MEFs, reverse transcribed and analyzed by real-time qPCR with primers that spanned exons 6 to 7 or 13 to 14 (Table 3). To quantitate hedgehog signaling, mRNA was

**Table 3. qPCR primer sequences used in this study.**

| Primers | Accession | Sequence | Tm | Size |
|---|---|---|---|---|
| MmGapdhExon3for | NM_008084 | GCAATGCATCCTGCACCACCA | 61.1 | |
| MmGapdhExon4rev | | TTCCAGAGGGGCCATCCACA | 61.1 | 138 |
| MmGli1Exon4for | NM_010296 | CCAGGGTTATGGAGCAGCCAGA | 61.2 | |
| MmGli1Exon5rev | | CTGGCATCAGAAAGGGGCGAGA | 61.5 | 135 |
| MmIft140exon6for | NM_134126 | GGAGAGAGGCACTGGTTGTGGTCA | 62.6 | |
| MmIft140exon7rev | | GGCAGCCTGTCTTCCCACTCAACT | 63.1 | 117 |
| MmIft140exon13for | NM_134126 | ACACCGTGGAGCCAAACCGACT | 63.4 | |
| MmIft140exon14rev | | TCCCACAGACGTCCAGGAAGCA | 62.6 | 111 |

isolated from cells that were treated or not treated with SAG for 24 h. After reverse transcribing, the levels of *Gli1* and *Gapdh* message were determined by real-time qPCR (Table 3).

## Protein analysis

Embryos were dissected and dispersed in denaturing gel loading buffer (10 mM Tris (pH 8.0), 16 mM dithiothreitol, 1 mM ethylenediaminetetraacetic acid, 5% sucrose, 1% sodium lauryl sulfate), passed through a 22 gauge needle to break up DNA, heated at 95 degrees C for 5 min, and separated by polyacrylamide gel electrophoresis. Proteins were transferred to Immobilon-P membrane (Merck Millipore) and westerns preformed with Ift140 [23] and γ-tubulin (GTU-88, Sigma) antibodies. Western blots were developed by chemiluminescence (Super Signal West Dura, Pierce Thermo) and imaged using an LAS-3000 imaging system (Fujifilm) or film. Anti-MmIFT140 was made by expressing the last 356 residues of the mouse protein in bacteria as a maltose-binding protein fusion and injecting into rabbits. Antibodies were affinity-purified against the same fragment expressed as a glutathione S-transferase fusion [23].

## Statistics

Statistical results were obtained from at least 3 independent experiments. Statistical differences between groups were tested by *t* tests, one-way ANOVA, two-way ANOVA, or Chi square as described in the figure legends. Differences between groups were considered statistically significant if $p < 0.05$. Otherwise, nonsignificant (n.s.) was labeled. Statistical significance is denoted with asterisks (* $p < 0.05$; ** $p < 0.01$; *** $p < 0.001$, **** $p < 0.0001$). Error bars indicate standard deviation (SD).

## Supporting information

**S1 Table. Midgestation lethality of Ift140[null1/null1] embryo.** *Subset of -/- embryos that died (no heartbeat or blood flow, or already undergoing resorption).
(DOCX)

**S2 Table. Summary of Cre lines used in this study.** [†]Mouse Genome Informatics (MGI) https://www.informatics.jax.org/.
(DOCX)

**S3 Table. Summary of mouse breeding used to generate embryos analysed in this study.** [†]When using tamoxifen-inducible Cre (*CAGGCre-ER*, *Foxa2Cre-ER*), tamoxifen was administered to pregnant mothers at embryonic ages listed.
(DOCX)

**S1 Fig. Scanning EM of E9 hearts.** Scanning electron microscopy reveals heart looping defects in *Ift140$^{null1/null1}$* embryos. Scale bar is 0.5 mm.
(TIF)

**S2 Fig. Loss of IFT140 after tamoxifen treatment.** (A) Western blot showing IFT140 levels in whole embryo lysates (*Ift140$^{flox/+}$ CAGGCre-ER$^+$* vs. *Ift140$^{flox/null1}$ CAGGCre-ER$^+$*) 48 h after treatment of the mother with 0.1 ml (1 mg) tamoxifen administered by oral gavage. Embryos were treated at E9 and harvested at E11. γ-tubulin is a loading control. (B) Quantification of the extent of IFT140 reduction 48 h after treatment of the mother with tamoxifen. Level of IFT140 was normalized between embryos using γ-tubulin and then experimental and control embryos within a litter were ratioed with controls set to 100%. Raw counts were normalized to controls from the same litter. ***$p < 0.0001$, unpaired Student *t* test. Error bar is standard deviation. The data underlying this figure can be found in Supporting information S1 Data.
(TIF)

**S3 Fig. *Mef2c-Cre, Tie2-Cre*, or tamoxifen-driven *Foxa2Cre-ER* deletion of *Ift140* does not cause extensive cardiac phenotypes.** (A–D) Deletion of *Ift140* by *Mef2c-Cre* or by tamoxifen-induced *Foxa2Cre-ER* (tamoxifen administered at E6.5, E7.5, or E8.5) display normal whole body gross anatomy. (E–J) Deletion of *Ift140* by *Mef2c-Cre*, *Tie2-Cre*, or by tamoxifen-driven *Foxa2Cre-ER* did not affect cardiac and great vessel anatomy. (K, L) Deletion of *Ift140* by *Tie2-Cre* results in eyelid closure defects and supernumerary mammary glands (arrow). LV: left ventricle; cx: cerebral cortex; sc: spinal cord; mb: midbrain; fl: forelimb; hl: hindlimb; e: eye; op: otic placode; hf: hair follicles; lv: liver; ln: lungs; i: small intestine; A: aorta; P: pulmonary trunk; LV: left ventricle; RV: right ventricle; LA: left atria; RA: right atria. Scales bars: (A–D) = 1 mm, (E–J) = 0. 5 mm.
(TIF)

**S4 Fig. Tamoxifen treatment appears to cause small ventricular septal defects.** (A–C) A small number of littermate controls (*Ift140$^{+/+}$*) collected from tamoxifen treated litters were found to have small VSDs. (D–F) Similar small ventricular septal defects were also seen in a small number of embryos with tamoxifen-driven Cre-specific knockdown, including *Foxa2Cre-ER* (D, E) and *Mef2c-Cre* (F). As these defects were seen across both wild type and experimental knockdown groups, they were excluded from phenotypic analysis and categorized as possible experimental artifacts. A: aorta; P: pulmonary trunk; LV: left ventricle; RV: right ventricle; LA: left atria; RA: right atria; VSD: ventricular septal defect; LSVC: left superior vena cava. All scales bars = 0.5 mm.
(TIF)

**S5 Fig. Schematic representation of genetically modified *Ift140* genes used in study.** Numbered black boxes indicate exon protein-coding regions; arrows: LoxP sites; ATG: start codon, Frt: flippase recombination target recognition site; LacZ: beta-galactosidase gene; Neo: neomycin-resistance gene.
(TIF)

**S1 Raw Images. Full size western blots. Top blots:** Left membrane was probed for Ift140 while the right was probed for gamma tubulin. The lanes used in Fig 4B are in the white box. **Middle blot:** Membrane was cut (arrow) and the top half probed for Ift140 and the lower half probed for gamma tubulin. The lanes used in Fig 4B are in the white box. **Bottom blot:** Membrane was cut (arrow) and the top half probed for Ift140 and the lower half probed for gamma tubulin. The left image is the western blot superimposed on an image of the membrane while

the right image is only the western blot. The lanes used in S2 Fig are in the boxes.
(TIF)

**S1 Data. Excel sheet containing the raw data used to generate all graphs found in the main figures.** Namely: Figs 2M, 3N, 4A, 5A, and S2B.
(XLSX)

**S1 Movie. 3D reconstruction of a wild-type E16.5 embryo processed using episcopic confocal microscopy highlighting normal cardiac anatomy.** Ao: Aorta, dAo: Descending aorta, DA: Ductus arteriosus, LA: Left atria, LV: Left ventricle, MV: Mitral valve, PA: Pulmonary artery, PT: Pulmonary trunk, RA: Right atria, RV Right ventricle, TV: Tricuspid valve.
(MP4)

**S2 Movie. 3D reconstruction of an E16.5 *Ift140$^{null1/null1}$* embryo processed using episcopic confocal microscopy highlighting abnormal cardiac anatomy.** AVSD: Atrioventricular septal defect, CA: Common atria, L: Liver, LA: Left atria, LV: Left ventricle, PTA: Persistent truncus arteriosus, RA: Right atria, RV: Right ventricle, SC: Spinal cord.
(MP4)

**S3 Movie. 3D reconstruction of a wild-type E16.5 embryo processed using episcopic confocal microscopy highlighting normal cardiac outflow tact development.** Ao: Aorta, dAo: Descending aorta, ASLV: Aortic semilunar valve, DA: Ductus arteriosus, LA: Left atria, LB: Left Bronchus, LPA: Left pulmonary artery, LV: Left ventricle, PA: Pulmonary artery, PSLV: Pulmonary semilunar valve, PT: Pulmonary trunk, RA: Right atria, RB: Right Bronchus, RCA: Right carotid artery, RPA: Right pulmonary artery, RV Right ventricle, T: Trachea.
(MP4)

**S4 Movie. 3D reconstruction of an E16.5 *Ift140$^{null1/null1}$* embryo processed using episcopic confocal microscopy highlighting abnormal cardiac outflow tact development.** dAo: Descending aorta, AVSD: Atrioventricular septal defect, CA: Common atria, E: Esophagus, IVC: Inferior vena cava, L: Liver, PTA: Persistent truncus arteriosus, RV: Right ventricle, S: Stomach, TEF: Tracheoesophageal fistula, UL: Underdeveloped Lung.
(MP4)

**S5 Movie. 3D reconstruction of a wild-type E16.5 embryo processed using episcopic confocal microscopy highlighting normal Trachea/Esophagus development.** Ao: Aorta, dAo: Descending aorta, E: Esophagus, LA: Left atria, LB: Left Bronchus, LCA: Left carotid artery, LV: Left ventricle, MV: Mitral valve, PV: Pulmonary vein, RA: Right atria, RB: Right Bronchus, RCA: Right carotid artery, RV: Right ventricle, S: Stomach, SCV: Subclavian vein, T: Trachea, TV: Tricuspid valve, VC: Vena cava.
(MP4)

**S6 Movie. 3D reconstruction of an E16.5 *Ift140$^{null1/null1}$* embryo processed using episcopic confocal microscopy highlighting Tracheoesophageal fistula.** E: Esophagus, IVC: Inferior vena cava, L: Liver, S: Stomach, SC: Spinal cord, SCV: Subclavian vein, SVC: Superior vena cava, TEF: Tracheoesophageal fistula, UL: Underdeveloped Lung.
(MP4)

**S7 Movie. 3D reconstruction of a wild-type E16.5 embryo processed using episcopic confocal microscopy highlighting normal chest/lung development.** dAo: Descending aorta, D: Diaphragm, E: Esophagus, L: Liver, LB: Left Bronchus, LL: Left lung lobe, RB: Right Bronchus, RL(SL): Right lung (Superior lobe), RL(ML): Right lung (Middle lobe), RL(IL): Right lung

(Inferior lobe), RL(PCL): Right lung (Post-caval lobe).
(MP4)

**S8 Movie. 3D reconstruction of an E16.5 *Ift140^{null1/null1}* embryo processed using episcopic confocal microscopy highlighting abnormal chest/lung development.** dAo: Descending aorta, L: Liver, SCV: Subclavian vein, SVC: Superior vena cava, TEF: Tracheoesophageal fistula, UL: Underdeveloped Lung.
(MP4)

## Acknowledgments

We thank Drs. S. Jones (UMass Chan Transgenic Mouse Core) and G. Hendricks (UMass Chan Electron Microscopy Core) for assistance during this work.

## Author Contributions

**Conceptualization:** Richard J. B. Francis, Cecilia W. Lo, Gregory J. Pazour.

**Data curation:** Cecilia W. Lo, Gregory J. Pazour.

**Formal analysis:** Richard J. B. Francis, Jovenal T. San Agustin, Heather L. Szabo Rogers, Cheng Cui, Julie A. Jonassen, Cecilia W. Lo, Gregory J. Pazour.

**Funding acquisition:** Cecilia W. Lo, Gregory J. Pazour.

**Investigation:** Richard J. B. Francis, Jovenal T. San Agustin, Cheng Cui, Julie A. Jonassen, Thibaut Eguether, John A. Follit.

**Methodology:** Richard J. B. Francis, Cecilia W. Lo, Gregory J. Pazour.

**Project administration:** Cecilia W. Lo, Gregory J. Pazour.

**Resources:** Cecilia W. Lo, Gregory J. Pazour.

**Supervision:** Cecilia W. Lo, Gregory J. Pazour.

**Validation:** Heather L. Szabo Rogers.

**Visualization:** Richard J. B. Francis.

**Writing – original draft:** Richard J. B. Francis, Cecilia W. Lo, Gregory J. Pazour.

**Writing – review & editing:** Richard J. B. Francis, Julie A. Jonassen, Thibaut Eguether, John A. Follit, Cecilia W. Lo, Gregory J. Pazour.

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
