## [Editor Report · Decision Letter 0]

3 Jul 2023

Dear Cecilia, 

Thank you for submitting your manuscript entitled "Autonomous and non-cell autonomous etiology of ciliopathy associated structural birth defects" for consideration as a Research Article by PLOS Biology.

Your manuscript has now been evaluated by the PLOS Biology editorial staff as well as by an academic editor with relevant expertise and I am writing to let you know that we would like to send your submission out for external peer review.

Once your full submission is complete, your paper will undergo a series of checks in preparation for peer review. After your manuscript has passed the checks it will be sent out for review. To provide the metadata for your submission, please Login to Editorial Manager (https://www.editorialmanager.com/pbiology) within two working days, i.e. by Jul 06 2023 11:59PM.

Kind regards,

Lucas

Lucas Smith, Ph.D.

Senior Editor

PLOS Biology

lsmith@plos.org

---

## [Decision Letter · Decision Letter 1]

4 Oct 2023

Dear Dr Francis,

Thank you for your patience while your manuscript "Autonomous and non-cell autonomous etiology of ciliopathy associated structural birth defects" was peer-reviewed at PLOS Biology. It has now been evaluated by the PLOS Biology editors, an Academic Editor with relevant expertise, and by several independent reviewers. 

In light of the reviews, which you will find at the end of this email, we would like to invite you to revise the work to thoroughly address the reviewers' reports.

As you will see below, the reviewers are largely positive about the study. However they have identified a number of important issues and we think these will need to be carefully addressed before we can consider your manuscript for publication. While many of these comments can be addressed with clarifications or textual changes, we encourage you to quantify the extent of cilia loss when embryos are treated at different times with tamoxifen and to clarify the level of Shh in neural crest cells, as reviewer 1 suggests. We think that adding this data will strengthen the study. 

Given the extent of revision needed, we cannot make a decision about publication until we have seen the revised manuscript and your response to the reviewers' comments. Your revised manuscript is likely to be sent for further evaluation by all or a subset of the reviewers.

**IMPORTANT - SUBMITTING YOUR REVISION**

*Re-submission Checklist*

*Published Peer Review*

*PLOS Data Policy*

*Blot and Gel Data Policy*

Sincerely,

Lucas

Lucas Smith, Ph.D.

Senior Editor

PLOS Biology

lsmith@plos.org

REVIEWS

Reviewer #1: The authors have studied the phenotype of mutant mouse deficient in Ift40, and intraflagellar transport protein required for ciliogenesis. Their data suggest that Ift40 has a cell autonomous role in craniofacial formation and body wall closure. Un expectedly, the role of Ift40 in heart development was non-cell autonomous. These findings probably have general messages for our understanding of pathogenesis of ciliopathy. The data are in general convincing and nicely support the conclusions. The paper should be published, but it would make this paper more convincing if the following relatively minor points are addressed.

1)From previous studies with various cilia-less mutant mice such as Kif3a mutant, the first stage during development that requires cilia is E7.5-8.0 when node cilia play a role in left-right (L-R) patterning. As expected, the Ift40 null mutant exhibit L-R defects. In conditional deletion with CAG-CreER, however, administration of Tmx at E7.5 failed to induce L-R defects. This is most likely due to the time required for Cre-mediated deletion and to the stability of Ift40 protein. Thus, 19 % of Ift40 still remained 48 hours after Tmx treatment (Fig. S3). It was necessary to treat embryos with Tmx as early as E5.5, to see L-R defects. Tmx treatment at E6.5 resulted in L-R defects at a lower frequency. Are almost all node cilia lost in embryos treated at E5.5 and only a fewer cilia in those treated at E6.5? This can be easily examined by SEM.

2)The outflow tract (OFT) was severely affected in Ift40 null mutant and Tmx-treated conditional mutant embryos. On the other hand, The OFT remain nearly normal in Wnt1Cre conditional mutant. This was unexpected because the previous paper reported that Wnt1-Cre mediated deletion of Smo in neural crest cells result in OFT defects. Formally, the current data suggest that either Hedgehog signaling in neural crest cells was not downregulated by Wnt1Cre-mediated deletion or the OFT defects of Ift40 null mutant are not due to hedgehog deficiency in neural crest cells. Since this is an important finding of this paper, it can be clarified by examining the level of Shh level in neural crest cells with the Gli-LacZ. 

3) "Non-cell autonomous cilia mediated interactions between multiple cell lineages" (page 18): this phrase can be discussed in more detail. 

Reviewer #2: Francis et al. provide a very extensive and comprehensive analysis of the role of IFT140 during development using multiple mouse models. One of the many strengths was that the authors deleted Ift140 at different embryonic days (E5.5 vs. E7.5, E8.5), which revealed different roles for IFT140 at these developmental stages. The authors also use multiple cell-specific Cre's to determine the cell lineage that results in CHD. Surprisingly, none of these Cre's resulted in the CHD observed in the global knockouts, suggesting that deletion of Ift140 in multiple cell lineages causes the CHD. Images are beautiful and manuscript is very nicely written. 

Specific comments:

1. Could the authors please include the embryonic day at which each of the tissue-specific Cre's are expressed? 

2. Related to this, Fig. 11A shows a timeline at which deletion of Ift140 results in various phenotypes. In Fig. 11B, could the phenotypes also be shown relative to the timeline e.g. as alluded to by the authors, heart looping defects would not be expected in Cre's that are expressed after E5.5.

3. Fig 5E: Gli and Ptch1 expression are decreased, and Grem1 is increased. In the Cauli mouse mutant, Gli1, Ptch1 and Grem1 are increased leading to polydactyly. Could the authors please provide an explanation/rationale for the decreased Gli and Ptch1 in Ift140 mutants leading to polydactyly? Is the polydactyly preaxial e.g. duplication of digit 1?

4. In the discussion, could the authors please describe how they envision that CHD arises from deletion of Ift140 in multiple cell lineages. 

Reviewer #3: Autonomous and non-cell autonomous etiology of ciliopathy associated structural birth defects

Francis et al.

The work by Francis et al on an allelic series of the core IFT-A machinery component IFT140 in mouse is a huge undertaking to understand the astounding clinical pleiotropy and viable penetrance of features seen amongst ciliopathy patients. The constellation of features observed in patients with mutations in the IFT machinery are some of the most multisyndromic amongst the ciliopathies, underlining the fact that functional cilia are required on most cell types during development. This work by Francis et al uses mouse molecular genetics, advanced imaging and careful embryology to dissect the functional requirement for cilia during different stages and tissues during carefully choreographed mammalian heart development and how defects in these different lineages contribute to congenital heart defects (CHDs). The manuscript represents a massive amount of work with quality analysis and goes towards addressing important questions in unpicking the requirement for different cilia types during complex processes of organogenesis and morphogenesis. By addressing some of these points raised in review, I believe this work to be of high quality and interest to the broad readership of PLoS Biology.

Major points

1. Need a summary of the alleles and drivers used in this study- a schematic of where the null allele and where the point mutant 220 lie. Is this different than the Cauli allele mentioned line 111? Someway to schematically represent a lot of really complicated crosses and alleles. KO line- lines 136-137: truncated protein, but you show protein null. Consider aligning the narrative with lines 173. Similarly some figures, most of the episcopic confocal microscopy 3D or 2D are labelled p/d, and should be CAG-Cre? Null1 what about null2- do we need the number? It is confusing for readers. Nomenclature through all figures and text should be standardized- alleles should be in italics and title case for mouse, not all caps (i.e. Figure 3,6,7,8,9). (Formatting and nomenclature throughtout this manuscript need careful attention- but these text boxes allow neither italics or superscript frustratingly)

2. CRE excision variability, dynamics (protein pool turnover) and lack of CHD phenotype. This alluded to with the Foxa2mcm allele mosaic recombination and lack of heart phenotypes upon tamoxifen treatment. These experiments were not done with a reporter Z/AP or Z/EG mT/mG, and ciliation in target tissues was not looked at, is possible that cardiac phenotypes are masked by variability in recombination levels, there needs to be a caveat in the text.

3. Human ciliopathies- be consistent and maybe consider OMIM terms for the skeletal dysplasias you mention frequently. Sometimes its SRTD sometimes SRP, Sensenbrenner sometimes craniosyntosis- helps non-experts keep up.

Minor points

1. Nomenclature: Cre lines should be italicized throughout Tbx18-Cre or Wnt1-Cre (lines 43-44). Table 2 needs revising. Use the correct line nomenclature for clarity 'tamoxifen (Tmx)-inducible Cag-CreER' is this the one from the Macmahon lab, then it should be CAGGCre-ER (line 249). The FoxA2-Cre is the knock-in allele from the Moon lab, then it should be Foxa2mcm. Is the Gli1lz the knock-in reporter from the Joyner lab and the knock-in Ptch1lacZ from the Scott lab? For example line 469 uses two different nomenclatures for the same allele. If you need to explain the allele and don't want to do so in the text, include it in the methods. There are alternates to all of these and it is important to be able to clearly identify which one was used here. 

2. Nomenclature: reference to Hedgehog related processes in mammals needs attention. Mammalian proteins ALL CAPs, mouse genes Title case italics. Introduce as Hedgehog (Hh). i.e. GLI, Patched1 (Ptch1) gene/mRNA or PTCH1 for protein. (Lines 55-62).

3. All antibodies should be all caps IFT88 (line 180), PAX6, OLIG2, NKX6.1 (line 211). Typo in OLIG2 (not OLIGO2). Figure 4C,E- use the antibody name, not structure labelled.

4. Figure S2- control sections are not age or level matched? Wild type is earlier or more caudal? Induction of ventral populations change with time. Figure 5B- should NKX6.1 not be excluded from the FP in controls, only V3 and MN? Put in text line 207 what developmental stage was looked at?

5. Line 285/286- Clarify 'embryos collected at E12.5-14.5 from Tmx treatment at E7.5 had more severe phenotypes than than those collected at E16.5?' Why would this be the case- surely phenotypes would be consistent across developmental ages if treated at the same time, unless embryos are dying earlier.

6. Units on all blots should be kDa, not kD.

7. Line 337 should be Gli3Xt-j and craniosyntosis phenotypes are common in many syndromic ciliopathies affecting IFT.

8. Line 359: Can you separate lung underdevelopment from lack of a diaphragm and liver within the thoracic cavity? This is not what causes narrowed ribs in most SRTD/JTD patients, where bone growth is cause. This statement likely needs revision.

9. Line 382 requires a reference: 'IFT140 ciliopathy patients often are compound heterozygous, with one allele being null and a second allele with a missense mutation.'

10. Line 412 'precise role of Ift140 in hedgehog signaling may be context dependent.' COuldn't it also be to do partial reduction in IFT pools in your CRE deleted lines- hypomorphic versus null? 

11. Line 425 Omphalocoele is not neural crest defect people think of. It may be useful to expand a little that some skeletal muscles have neural crest contribution where long range migration of muscle precursors is required for lateral and body wall muscles needed for body wall closure. Please check this section.

12. Mouse breeding- lacking information on strain used.

13. Protein analysis- missing details of how protein lysates were made.

14. Line 820- instead of using the clone, use the antigen- acetylated tubulin?

15. Figure s3 legend- which Cre was used? What is loaded on A- lysates of which stage (representative to quantification below)? This could be clearer.

16. Figure S6- bottom label should reference Figure S3, Figure 4B- is gamma tubulin the loading control for the right blot- running of bands looks different and membrane looks cut? Labels on all say gamma tubulin and in legend but on blots labelled as alpha tubulin? Not specified in Fig 4 legend. Needs attention.

Edits 

1. CHD is plural- line 41. Surprisingly, CHD were not observed…

2. Line 48- Plural for ciliopathies- many are affected by CHD.

3. Line 82: italicize Ift140 - mutations are in genes.

4. Line 92: E13 is mid-gestation, not early.

5. Line 159: wildtype littermates. 

6. Line 180: do you mean centrioles instead of centrosomes?

7. Line 192: use micron symbol.

8. Line 247: Should read gain-of-function.

9. Line 263: Clarify: it reads 'embryonic lethality, necessitating earlier embryo collection at E11.5-E12.5'. Do you mean neonatal lethality or that mutant embryos were missing at birth? Clearer ways to say this.

10. Line 355: Clarify 'Both Ift140 mutants exhibited a similar wide array of SBD, even though IFT-A is generally considered less critical for ciliary assembly than IFT-B'. Cilia don't assemble properly if there is no IFT-A, as you have shown?

11. Line 444: should be short rib polydactyly (SRP).

12. Line 515: transcripts should be italics (see also Line 837)

13. Lines 768, 770: italicize Ift140220

14. Line 772: space between value and unit. (For all figure legends)

15. Line 774: italicize alleles in all figure legends

16. Line 895: craniofacial?

---

## [Editor Report · Decision Letter 2]

3 Nov 2023

Dear Dr Francis,

Thank you for your patience while we considered your revised manuscript "Autonomous and non-cell autonomous etiology of ciliopathy associated structural birth defects" for publication as a Research Article at PLOS Biology. This revised version of your manuscript has been evaluated by the PLOS Biology editors and the Academic Editor, who is completely satisfied with the revision. 

Based on the reviews on our Academic Editor's assessment of your revision, we are likely to accept this manuscript for publication. However, before we can editorially accept your paper, we need you to address a number of minor editorial points in a revision that we think will not take very long. 

**Please address the following editorial requests: 

1) TITLE: Given the public interest in this topic, we think the title of your piece should be edited to indicate that the study was done in mice. We therefore suggest it be changed to something like: 

""Autonomous and non-cell autonomous role of cilia in structural birth defects in mice""

2) ETHICS STATEMENT: Please update the ethics statement, in your materials and methods section, to include the specific national or international regulations/guidelines to which your animal care and use protocol adhered. Please note that institutional or accreditation organization guidelines (such as AAALAC) do not meet this requirement.

3) DATA: Thank you for providing the data underlying your figures as a supplemental file. Can you please update this to add the data from the supplemental figures as well (ex the underlying data for Supplemental Figure 2B)? Please also add a note to each relevant figure legend that directs readers to this file. You can add the sentence, "the data underlying this figure can be found in supplemental file RawDataForFigures.

4) BLOTS: Thanks also for providing the uncropped western blots related to this paper as a supplemental figure. A few requests regarding this data:

a. Rather than presenting these as a figure, can you provide them as a supplemental file, titled "S1_Raw_Images"? 

b. We ask that you please add a few more annotations to this file. Specifically, please indicate molecular weight markers. For bands that were not presented in the main figure, please add an 'x' over each irrelevant well (without obscuring the image). 

c. The blots provided for Fig 4B, Gamma tubulin seems to be slightly cropped. Please provide a less cropped version of this figure. If it is difficult to see the full membrane, please just provide the full page image generated by the scanner. 

5) CODE: Per journal policy, if any code was generated to support the conclusions of your manuscript, we would require that you make it available without restrictions upon publication. Please ensure that any code is sufficiently well documented and reusable, and that your Data Statement in the Editorial Manager submission system accurately describes where your code can be found.

We expect to receive your revised manuscript within two weeks. 

*Published Peer Review History*

*Press*

Sincerely,

Luke

Lucas Smith, Ph.D.

Senior Editor,

lsmith@plos.org,

PLOS Biology

---

## [Editor Report · Decision Letter 3]

9 Nov 2023

Dear Dr Francis,

Thank you for the submission of your revised Research Article "Autonomous and non-cell autonomous etiology of ciliopathy associated structural birth defects" for publication in PLOS Biology, and thank you for addressing our last editorial requests in this revision. On behalf of my colleagues and the Academic Editor, Caroline S Hill, I am pleased to say that we can in principle accept your manuscript for publication, provided you address any remaining formatting and reporting issues. These will be detailed in an email you should receive within 2-3 business days from our colleagues in the journal operations team; no action is required from you until then. Please note that we will not be able to formally accept your manuscript and schedule it for publication until you have completed any requested changes.

**IMPORTANT: As you address any formatting and reporting requests to come, we also ask that you attend to the following residual editorial requests: 

1) TITLE: Thanks for changing your title to "Autonomous and non-cell autonomous role of cilia in structural birth defects in mice". I see that this change was made in the 'track changes' version of the study - but does not seem to have made its way into the final version of the manuscript provided, or into our editorial manager system. Please make sure to update those before publication. 

2) DATA: Thank you for providing the underlying data related to Figure 2B - I noticed that the numerical values underlying this figure were added to the figure legend. Would you mind, instead, moving that data to a new tab in the RawDataForFigures file?

3) FIGURE S2: Please add annotations to the western blots presented in Figure S2A, indicating what samples were loaded in each well. 

4) FIGURE LEGENDS: I noticed that some of the figure legends lacked details about what statistics were performed. Can you please update the figure legends to include this information. (For example, details about statistics were missing from Fig S2 and Figure 5 figure legends). 

PRESS

Sincerely, 

Lucas Smith, Ph.D.

Senior Editor

PLOS Biology

lsmith@plos.org